# Delineation of homogenous regions using hydrological variables predicted by projection pursuit regression

Martin Durocher [1], Fateh Chebana [2], Taha B. M. J. Ouarda[2,3]

5  [1]Université du Québec à Trois-Rivières, University of Quebec,

3351, boul. des Forges, C.P. 500, Trois-Rivières, G9A 5H7, Canada

[2]Institut National de Recherche Scientifique (INRS-ETE), University of Quebec,

490 de la Couronne, Québec G1K 9A9, Canada

[3] Institute Center for Water Advanced Technology and Environmental Research (iWater),

Masdar Institue of Science and Technology, P.O. Box 54224, Abu Dhabi, UAE

*Corresponding author: Martin Durocher (martin.durocher@uqtr.ca)

**Abstract**

This study investigates the utilization of hydrological information in Regional Flood Frequency Analysis (RFFA) to enforce desired properties for a group of gauged stations. Neighborhoods are a particular type of regions that are centered on target locations. A challenge for using neighborhoods in RFFA is that hydrological information is not available at target locations and cannot be completely replaced by the available physiographical information. Instead of using the available
physiographic characteristics to define the center of a target location, this study proposes to introduce estimates of reference hydrological variables to ensure a better homogeneity. These reference variables represent nonlinear relations with the site characteristics obtained by projection pursuit regression; a nonparametric regression method. The resulting neighborhoods are investigated in combination with commonly used regional models: the index-flood model and regression-based models. The complete approach is illustrated on a real-world case study with gauged sites from the Southern part of the province of
Quebec, Canada, and is compared with the traditional approaches "Region of Influence" and "Canonical Correlation Analysis". The evaluation focuses on the neighborhood properties as well as prediction performances, with special attention to problematic stations. Results show clear improvements in neighborhood definitions and quantile estimates.

**Keywords**:  Index-flood model, Regional frequency analysis, Ungauged site, Region of influence, Projection pursuit
regression, Canonical Correlation Analysis.

## 1. Introduction

Accurate estimates of the risk of occurrence of extreme hydrological events are necessary for the minimization of the impacts of these events and for the optimal design and management of water resource systems. However, necessary information is not always available at the sites of interest. It is hence necessary to develop procedures to transfer, or to regionalize, the available information at existing gauged sites to the ungauged ones. Regional Flood Frequency Analysis (RFFA) represents a large class of techniques commonly used in water sciences to evaluate the risk of occurrence of extreme hydrological phenomena of rare magnitudes at ungauged locations (Haddad and Rahman, 2012; Hosking and Wallis, 1997; Laio et al., 2011; Pandey, 1998; Reis et al., 2005).

RFFA methods are usually composed of two main steps. The first step is the formation of homogenous regions. This step aims at pooling together sites that are approximately similar according to homogeneity criteria. Inside these homogenous regions, it is assumed that hydrological information can be reasonably transferred from gauged to ungauged locations (Cunnane, 1988). The second step, the estimation of flood quantiles, consists in the calibration of a regional model that characterizes the interrelation between hydrological variables of interest and explanatory physio-meteorological variables corresponding to known site characteristics. Consequently, RFFA is used to study unobserved hydrological behaviour from available hydrological and physio-meteorological information.

Neighborhoods are specific forms of regions that are not composed of a fixed set of stations, but are rather composed of gauged sites that are the most similar to a given target site. Hence, two distinct target locations will have their own distinct neighborhoods which may overlap. Comparative studies have shown that neighborhoods lead to better regional estimates than fixed regions (Burn, 1990; Ouarda et al., 2008; Tasker et al., 1996). To identify the most similar gauged sites in terms of hydrological properties, a notion of distance is needed. It allows to evaluate the proximity, or relevance, of each gauged site to the target location and to identify the most hydrologically similar gauged sites. However, when the target location is ungauged, this distance cannot be directly calculated due to the missing hydrological information. Physio-meteorological information is hence used for similarity evaluation. The traditional approach, based on the distance between site characteristics, is commonly referred to as the Region of Influence (ROI) model (Burn, 1990), which received a particular attention in the hydrological literature. The focus was mainly on the estimation of the model parameters, where for instance generalized least-squares were used to account for unequal variability in the at-site estimations (e.g. Griffis and Stedinger, 2007; Stedinger and Tasker, 1985) and to deal with the presence of spatial correlation (e.g. Kjeldsen and Jones, 2009).

Alternatively, Ouarda et al. (2001) used Canonical Correlation Analysis (CCA) to build neighborhoods from a canonical distance that accounts for the interrelation between flood quantiles and site characteristics. For this method, neighborhoods are formed by gauged sites that are the most similar to the target location, according to the distance between vectors of flood quantiles corresponding to different return periods. The CCA method in RFFA estimates the unavailable hydrological variables as linear combinations of site characteristics. Consequently, the available site characteristics are transformed into more meaningful "hydrological" quantities for the purpose of delineating neighborhoods. However, the CCA method suffers from some limitations, such as linearity and normality assumptions (He et al., 2011). Subsequent

studies have aimed at improving the CCA method by improving the CCA technique itself (Chebana and Ouarda, 2008; Ouali et al., 2015). However, little attention has been paid to the importance of properly choosing the hydrological quantities in the delineation step whereas much effort has been devoted to the modeling step. Indeed, Chebana and Ouarda (2008) employed an iterative linear procedure to estimate neighborhood centers and they showed that the quality of these centers' estimates is the crucial element to the improvement of the final model performance.

The present study aims to provide a general framework with more flexibility regarding the linearity and normality assumptions. This is achieved by replacing CCA in the prior analysis of hydrological variables by Projection Pursuit Regression (PPR), a nonparametric regression method recently considered as an estimation model in RFFA (Durocher et al., 2015). The present study is also interested in assessing the advantages of employing hydrological variables other than the at-site flood quantiles in prior modeling as well as considering a combination of these hydrological variables with site characteristics.

L-moments have already been used in RFFA to test the homogeneity of fixed regions when the target site is gauged (Chebana and Ouarda, 2007; Hosking and Wallis, 1997). In the present study, the prediction of the L-moments at ungauged sites is also considered to improve the delineation of the neighborhoods by reducing uncertainties. Moreover, a conceptual advantage of using L-moments conversely to at-site flood quantiles is that the L-moments do not depend on the subjective selection of at-site distributions.

The present paper is organized as follows. Section 2 presents the background material for the techniques used in the present research. Section 3 elaborates on the prior analysis of hydrological variables and their integration with the techniques presented in Section 2 to form a complete procedure. Section 3 suggests criteria for the evaluation of the predictive performances and the neighborhood properties. Section 4 illustrates the application of the method on a case study. Traditional ROI and CCA methods serve as references in order to evaluate the relative performance of the investigated method. Finally, concluding remarks are provided in the last section.

## 2. Background

### 2.1 Delineation of neighborhoods

In RFFA, neighborhoods are used to identify gauged sites from which information is transferred to the target location. A neighborhood is characterized by a center and a radius that delimits an area (not necessary in the geographical sense). Gauged sites inside the area delineate a region that includes relevant sites to the target location. At each site $i = 1, \ldots, n$, $p$ characteristics $\mathbf{x}_i = (x_{i,1}, \ldots, x_{i,p})$ are available. Typically, the ROI method forms neighborhoods according to a radius based on a metric $d$:

$$d(\mathbf{x}_i, \mathbf{x}_j) = \sqrt{\sum_{k=1}^{p} \frac{(x_{i,k} - x_{j,k})^2}{\sigma_k^2}} \tag{1}$$

where $\sigma_k$ is the standard deviation of $\{x_{i,k}\}_{i=1}^{n}$ the kth site characteristic (Eng et al., 2005).

Alternatively, CCA is a multivariate technique used to unveil the interrelation between two groups of variables. Let $Y$ and $X$ be normally distributed random vectors with zero means. The CCA method defines canonical pairs $(U_k, V_k)$ as linear combinations of the original random variables:

$$U_k = a_k X \tag{2}$$

$$V_k = b_k Y \tag{3}$$

where the correlations $\rho_k = corr(U_k, V_k)$ are sequentially maximal for $k = 1, \ldots, K$ under the conditions $corr(U_k, U_l) = corr(V_k, V_l) = 0$ for $k \neq l$. Only the canonical pairs $(U_k, V_k)$ with unit variances are considered.

To delineate neighborhoods, the CCA approach considers the canonical scores $\mathbf{u}_i = (a_1, \ldots, a_r)' \mathbf{x}_i$ and $\mathbf{v}_i = (b_1, \ldots, b_r)' \mathbf{y}_i$ that are respectively linear combinations of site characteristics $\mathbf{x}_i$ and flood quantiles corresponding to different return periods $\mathbf{y}_i$ for site $i$. Due to the missing hydrological information at the ungauged location denoted $i = 0$, the flood quantiles $\mathbf{y}_0$ and the corresponding linear combination $\mathbf{v}_0$ are unknown. Nevertheless, CCA provides a linear estimate $\mathbf{v}_0 \approx \Lambda \mathbf{u}_0$, where $\Lambda = \mathrm{diag}(\rho_1, \ldots, \rho_K)$. Accordingly, a neighborhood is delineated in the canonical space based on the distance:

$$d(\mathbf{v}_i, \Lambda \mathbf{u}_0) = (\mathbf{v}_i - \Lambda \mathbf{u}_0)' (I - \Lambda^2)^{-1} (\mathbf{v}_i - \Lambda \mathbf{u}_0) \tag{4}$$

More details on the CCA approach in RFFA can be obtained in Ouarda et al. (2001) and Ouarda (2016).

## 2.2 Multiple regression

In RFFA, two types of regional models are often considered to predict flood quantiles corresponding to given return periods: the index-flood model and the regression-based model (Ouarda et al., 2008). The index-flood model predicts a target distribution by assuming that all distributions inside the region are proportional to a regional distribution, up to a scale factor called index-flood. The flood quantile of interest at a target location is then calculated from the regional distribution based on the predicted index-flood (e.g., Chebana and Ouarda, 2009; Dalrymple, 1960; Stedinger and Lu, 1995). Conversely, the regression-based model considers directly the at-site estimates of the desired flood quantiles for prediction. Flood quantiles are then predicted at their target locations by the regression equations estimated within the neighborhoods (Pandey and Nguyen, 1999).

Even though they proceed differently, both the index-flood model and the regression-based model may use the same multiple regression techniques to transfer information to an ungauged location. For the sake of simplicity, the term hydrological variables is used to designate the corresponding output variables $z_i$ of these models at location $i = 1, \ldots, n$. Consequently, for the index-flood model, $z_i$ is the index flood, while for a regression-based model the hydrological

variable $z_i$ is the flood quantile corresponding to the return period of interest.

Multiple regression models assume linear interrelation between the hydrological variable $z_i$ and the site characteristics $\mathbf{x}_i$. Consequently, in several cases, transformations are necessary to meet this assumption. For instance, the power law form is frequently used to model flood quantiles:

$$z_i = e^{\beta_0} \times x_{i,1}^{\beta_1} \times \ldots \times x_{i,p}^{\beta_p} \times \varepsilon_i \qquad (5)$$

where $\beta' = \left( \beta_0, \beta_1, \ldots, \beta_p \right)$ are parameters and $\varepsilon_i$ is an error term. Applying a logarithmic transformation is sufficient to cast Eq. (5) into a linear model. In general, a proper transformation is assumed for the hydrological variables $y_i = g\left( z_i \right)$ being linearly related to the site characteristics.

In line with previous notations, let $\mathbf{y} = \left( y_1, \ldots, y_n \right)$ be the hydrological variables, $\mathbf{X}$ be the design matrix of the site

characteristics $x_{i,j}$ with intercept, and $\varepsilon = \left( \varepsilon_1, \ldots, \varepsilon_n \right)$ be the error term. Hence in matrix notation, a multiple regression model has the form:

$$\mathbf{y} = \mathbf{X}\beta + \varepsilon \qquad (6)$$

and according to the least-squares theory, the estimates of the parameters are:

$$\hat{\beta} = \left( \mathbf{X'X} \right)^{-1} \mathbf{X'y} \qquad (7)$$

**2.3 Projection pursuit regression**

Some methods predict hydrological variables without the formation of regions, such as physiographical kriging (Castiglioni et al., 2009; Chokmani and Ouarda, 2004), generalized additive models (Chebana et al., 2014) and artificial neural networks (Dawson et al., 2006; Ouarda and Shu, 2009). More recently, Projection Pursuit Regression (PPR) was introduced to provide a flexible nonparametric regression approach to describe the nonlinearity that is present in the

relationship between hydrological variables and site characteristics. PPR was used in the RFFA context by Durocher et al. (2015) to directly predict flood quantiles without delineation.

The basic elements of a PPR model are $k = 1, \ldots, m$ functions $f_k$ called terms and defined as:

$$f_k\left( \mathbf{X} \right) = g_k\left( \alpha_k' \mathbf{X} \right) \qquad (8)$$

where directions $\alpha_k$ are vectors of coefficients and $g_k$ are smooth functions. The directions $\alpha_k$ are coefficients that

respect $|\alpha| = 1$ and determine a predictor $\alpha_k' \mathbf{X}$ as relevant linear combinations of the site characteristics $\mathbf{X}$. The terms are then combined into a regression model:

$$\mathbf{y} = \mu + \sum_{k=1}^{m} f_k(\mathbf{X}) + \varepsilon \tag{9}$$

where $\mu$ is the global mean and $\varepsilon$ is an error term. Notice that the orthogonality between directions $\alpha_k$ is not imposed, hence the predictors $\alpha_k'\mathbf{X}$ and $\alpha_l'\mathbf{X}$ for $k \neq l$ may be correlated. Consequently, PPR allows for the interaction between site characteristics, which leads to a large variety of regression models (Hastie et al., 2009).

The components $\alpha_k$ and $g_k$ of the model in Eq. (9) are estimated by the least-squares approach (Friedman et al., 1983). For a unique direction ($m = 1$), PPR can be estimated by standard nonlinear algorithms (Yu and Ruppert, 2002), but in general a stagewise algorithm is adopted to find a proper solution (Friedman and Tukey, 1974). Comparative studies show that PPR has a predictive performance that is similar to artificial neural networks (Bishop, 1995; Hwang et al., 1994). However, Durocher et al. (2015) indicated that in RFFA, PPR reduces to more parsimonious models than artificial neural

networks, which provides an explicit expression of the regression equations.

## 3 Methodology

The present study deals with neighborhood delineation and focuses more precisely on the identification of reliable estimates of the hydrological centers of these neighborhoods. For the sake of simplicity, the variables forming these centers will be referred to as reference variables, because they represent the reference to evaluate the similarity between a target

location and the gauged sites. Reference variables can take different forms, such as site characteristics, hydrological variables or a combination of both. Their nature is important, because it determines the properties that are deemed to be important between close sites. The particularity of the present method is that PPR can be used to predict these neighborhood centers (prior to the RFFA modeling step) when some of the reference variables are unknown hydrological variables. Accordingly, the proposed method will be referred to as RVN for Reference Variable Neighborhoods.

### 3.1 Estimation of the reference variables

The general procedure can be described by the steps below:

1.  Select the reference variables

2.  If necessary, predict the reference variables that are not available at the target site

3.  Compute the distance between sites

4.  Form the neighborhood based on the previous distance

5.  Fit a regional model on the neighborhood

6.  Predict the target site and evaluate a performance criterion

In step 1, the selection of a set of the reference variables can be subjective and depends on the problem at hand. In the present study, the backward stepwise selection procedure is considered to remove from an initial set of references variables

those that do not contribute to the prediction power of the model. This selection procedure is more objective and depends on a performance criterion. In the present study the relative root mean square error (RRMSE) criterion is chosen for this purpose and will be described in section 3.2. The backward stepwise selection is illustrated in Figure 1 and consists in removing in turn each reference variable temporarily from the model and performing the remaining steps (2-6) in order to compute the RRMSE. Therefore, the reference variable whose removal leads to the best RRMSE, is permanently removed. The process is repeated until all reference variables cannot be removed without altering the RRMSE.

Step 2 is required only if some reference variables are unknown at the target sites. Otherwise, if we designate the target location by $i = 0$, the radius of the neighborhood used in step 3 can be computed as $h_i = d(\mathbf{t}_i, \mathbf{t}_0)$ where $d$ is a metric and $\mathbf{t}_i' = (t_{i,1}, \ldots, t_{i,q})$ are the reference variables of the $i$th site. For simplicity, the Euclidian metric $d$ is considered throughout the present study, but other metrics or dissimilarity measures can be employed as well. In particular, the Mahalanobis distance, the weighted distance or the depth functions could be considered (Chebana and Ouarda, 2008; Cunderlik and Burn, 2006; Ouarda et al., 2000).

If certain hydrological information is unavailable at the target location, the estimation of the hydrological reference variables is necessary to produce an estimate $\mathbf{t}_0 = f(\mathbf{x}_0)$ in step 2 from site characteristics $\mathbf{x}_0$ at the target location. This substitution leads in step 3 to the distance $h_{(i)} = d[\mathbf{t}_i, f(\mathbf{x}_0)]$, which may be seen as an approximation of the true distance $h_i$. This study considers PPR models in order to fit every hydrological reference variable as described in section 2.3. The motivations for adopting PPR are that it does not require a prior delineation of regions, it accounts for nonlinear relationships, it has good predictive performances and it leads to a straightforward interpretation of the reference variables when a few directions $\alpha_k$ are necessary (Durocher et al., 2015).

If the hydrological variables $\mathbf{t}_0$ were known at the target location, the distance $h_i$ would be available and the neighborhood that truly regroups the most hydrologically similar sites to the target location can be identified. However, in practice this true neighborhood is unknown. Using instead the estimate $f(\mathbf{x}_0)$ has as effect that some sites are falsely suggested as more hydrologically similar than other sites. Figure 2 illustrates a region with several sites where two neighborhoods are resulting from the RVN method with different predicted centers. The target site is illustrated as a green filled circle and the neighborhood is formed by the 10 nearest sites indicated by small empty circles. The other sites are designated by crosses. The red and blue neighborhoods are delineated by circles where the radius is selected to include the 10 nearest sites. The predicted center of the red neighborhood is closer to the target site. Consequently, it can be seen that except from one site, the same sites as the target neighborhood are included (empty circles). On the other hand, the blue neighborhood has a predicted center that is located further from the target site and hence a lower proportion of the sites truly closer to the target are found. This shows the importance of correctly predicting the neighborhood centers in order to identify sites that are truly similar to the target site.

The errors related to prediction of the hydrological reference variables suggest that the RVN method may include an additional source of uncertainty. Indeed, the same source of uncertainty is present among the sites of a neighborhood

delineated on the basis of the site characteristics (*i.e.* that the average of the hydrological variables in the neighborhood is not a perfect predictor). This could be seen as an advantage of the RVN method since it directly assesses this source of uncertainty and tries to reduce it.

Step 1-3 are the particularity of the RVN method, while the other steps are common in RFFA and are explained in section 2. In the remainder of this study, step 4 uses a specific type of neighborhoods that is composed of a fixed number of the nearest sites (Eng et al., 2005; Tasker et al., 1996), but could also be constrained to the degree of the homogeneity of the neighborhoods (Ouarda et al., 2001). Consequently, the selected gauged sites can be obtained by sorting $h_{(i)}$ and keeping the desired number of sites. Notice that even though $h_{(i)}$ does not exactly approximate $h_i$, both distances will lead to the same neighborhoods if they preserve the ranks. Finally, step 5 consists in the estimation of the flood quantiles using either the index-flood or the regression-based model.

Notice that the RVN method may be seen as a generalization of the ROI and the CCA methods in RFFA. Indeed, the ROI method corresponds to the RVN method for which all the reference variables are site characteristics. In that case, $\mathbf{t}_0 = f(\mathbf{x}_0)$ is known and PPR is not necessary in step 2. Similarly, the CCA approach may be seen as the special case for which the reference variables are the canonical pairs in Eq. (4) and CCA is used, instead of PPR, to predict them in step 2.

## 3.2 Evaluation criteria

For the RVN method presented above, the neighborhood sizes must be calibrated according to an objective criterion. In this regards, the leave-one-out cross-validation approach is a general strategy to assess the performance of the predicted hydrological variables $z_i$ at site $i = 1, \ldots, n$. In turn, each gauged site $i$ is considered as an ungauged target location. From the remaining gauged sites, predicted values $z_{(i)}$ can be obtained without using the hydrological information at the target location. Discrepancies between the sampled and the predicted values are used to define evaluation criteria. Notice that the hydrological variables are transformed $y_i = g(z_i)$. Hence, if $\overline{y}$ is the sample mean of the $y_i$, then an appropriate global performance measure is the Nash-Sutcliffe criterion:

$$\text{NHS} = 1 - \frac{\sum_{i=1}^{n} \left[ y_i - y_{(i)} \right]^2}{\sum_{i=1}^{n} \left[ y_i - \overline{y} \right]^2} \tag{10}$$

Additionally, the predictive performance is examined at the original scale by the RRMSE:

$$\text{RRMSE} = \sqrt{\frac{1}{n} \sum_{i=1}^{n} \left( 1 - \frac{z_{(i)}}{z_i} \right)^2} \tag{11}$$

The choice of the reference variables is an important aspect and a set of reference variables should be chosen in order to enforce the desired properties. For instance, with the index-flood model the assumption of a regional distribution suggests

that, apart from the index-flood, the at-site distributions must be proportional to a regional distribution. A heterogeneity measure based on the dispersion of the L-coefficient of variation (LCV) is shown to be a proper way to ensure that the LCV is relatively constant (Viglione et al., 2007). Accordingly, let $I_j$ be the set of indices for the $N$ nearest gauged sites to the target location $j$ during the cross-validation process. The regional LCV $\hat{\theta}_{(j)}$ is calculated as the average:

$$\hat{\theta}_{(j)} = \frac{1}{N} \sum_{i \in I_j} \theta_i \tag{12}$$

of the at-site LCV $\theta_i$ inside the jth region. The heterogeneity measure is defined as:

$$H_{(j)} = \sum_{i \in I_j} \left( \theta_i - \hat{\theta}_{(j)} \right)^2 \tag{13}$$

In their procedure, Hosking and Wallis (1997) used this heterogeneity measure to test for regional homogeneity, which implies that the regional LCV can be considered constant. Hence, the result of this test allows deciding if a region must be divided into smaller and more homogenous sub-regions. In the present study the size of the neighborhoods is the same. Hence, if a homogeneity test is performed with a given neighborhood size, some of the neighborhoods will be considered homogenous, while the others will be considered heterogeneous (Das and Cunnane, 2010). However, the heterogeneity measure in Eq. (13) remains a useful indicator of dispersion for the regional LCV $\hat{\theta}_{(j)}$ inside a neighborhood. Consequently, a smaller $H_{(j)}$ suggests that the regional LCV $\hat{\theta}_{(j)}$ is measured with less uncertainty.

15      To facilitate the interpretation of the results and to ensure the comparability between neighborhoods, the heterogeneity measure $H_{(j)}/N$ is considered instead. The measure represents the sample variance of the LCV for the jth target location. This heterogeneity measure is standardized by $H/n$, where $H$ is the heterogeneity measure in Eq. (13) calculated on all $n$ available gauged sites. The resulting ratio corresponds to a scale-free heterogeneity measure, where a value under one provides evidence of a less heterogeneous neighborhood in comparison to the whole dataset. Therefore, the Average Heterogeneity Measure (AHM) criterion below is defined as the average of every neighborhood considered in the cross-validation process:

$$\mathrm{AHM} = \frac{1}{N \cdot H} \sum_{j=1}^{n} H_{(j)} \tag{14}$$

This criterion is not specific to a given target location, but represents the global level of heterogeneity resulting from a given delineation method, such as ROI, CCA or RVN. In particular, a delineation method with a smaller AHM is an indication that, on average, a more precise regional LCV is used to predict flood quantiles.

Another desired property for a neighborhood is that it leads to estimation models with less uncertainty. For the index-flood model, this implies in particular less uncertainty in the prediction of the index-flood, while for regression-based models, it implies less uncertainty in the prediction of flood quantiles. For a multiple regression model, the uncertainty can

be quantified by the residual variance:

$$s^2_{(j)} = \frac{1}{N} \sum_{i \in I_j} \left( e_{i,(j)} \right)^2 \qquad (15)$$

where $e_{i,(j)}$ is the residual at the ith gauged site, when predicting the jth target location in the cross-validation process. Notice that a regression model fitted on two different neighborhoods (for the same target location) can obtain identical values, but lead to different levels of uncertainty. In this study, a neighborhood with a smaller residual variance is said to be relatively more efficient.

During the cross-validation process, the sample variance of the regression models can be calculated for every site, which leads to the Average Relative Efficiency (ARE) criterion defined by:

$$\text{ARE} = \frac{1}{ns^2} \sum_{j=1}^{n} s^2_{(j)} \qquad (16)$$

where the residual variance $s^2$ is calculated from the multiple regression model on the whole dataset. This criterion is similar to the AHM criterion as is standardized to a scale-free measure. This criterion can be used to identify the delineation method which achieves, on average, the smallest residual variances for each neighborhood. The ARE and the AHM criteria are used in the present study, along with the NHS and RRMSE to access the performances of the various models.

## 4. Application

### 4.1 Data

To validate the RVN method, RFFA is carried out in a real-world case study using both the index-flood model and the regression-based model. The hydrological variables of interest are the flood quantiles corresponding to a return period of 100 years, denoted Q100. The analysis is performed on 151 sites located in the southern part of the Province of Quebec, Canada. Figure 3 illustrates the location of these sites. Each site has at least 15 years of data, and the average record length is 31 years. The usual hypotheses of stationarity, homogeneity and independence are verified for all 151 data series. Only a brief description of the data and the at-site frequency analysis is provided since the elements were already presented in details in previous studies (e.g., Chokmani and Ouarda, 2004).

The at-site distributions are selected among several families including: generalized extreme values (GEV), Pearson type III (P3), generalized logistic (GLO) and log-normal with 3 parameters (LN3). In general, the estimation of the at-site distribution was achieved by maximum likelihood and the final choices of distributions are based on the Akaike information criterion. Recent studies on the same dataset have identified 4 relevant site characteristics (Chebana et al., 2014; Durocher et al., 2015), which are used in the present analysis: the drainage area or BV ($km^2$), the fraction of the basin area occupied by lakes or PLAC (%), the annual mean liquid precipitation or PLMA (mm) and the longitude or LON. Proper transformations are applied on these site characteristics in order to obtain approximately standard normal distributions.

**4.2 Determination of the neighborhood centers**

Steps 1-2 of the RVN method represent the selection of the reference variables and, if necessary, the estimation of the hydrological reference variables at the target locations. Two initial groups of reference variables are considered and updated by backward stepwise selection. The first group is based on L-moments only and the second is based on the combination of L-moments and site-characteristics. The acronym LM for L-moment and HYB for Hybrid are used to identify the two groups. More precisely, the L-moments considered for both groups are the sample average (L1), the LCV, the L-coefficient of skewness (LSK) and the L-coefficient of kurtosis (LKT). These reference variables are transformed and standardized to obtain zero mean and unit variance. The transformation used for L1 and LCV is the logarithm, while for LSK and LKT, the transformation is $\log(x - m_x + 1)$, where $m_x$ is the minimum of the reference variables.

A specific implementation of PPR is assumed, which considers the smooth functions $g_k$ in Eq. (8) as cubic spline polynomials with 5 equally spaced knots. The number of knots is validated by cross-validation using the NHS criterion. Notice that for the fitting of LSK, one site has a very low standardized residual of approximately -6. Consequently, this site is considered as an outlier and removed from the estimation of the reference variables. In previous studies (e.g., Chokmani and Ouarda, 2004), this site was identified as one of a few problematic sites that are difficult to predict due to an underestimated drainage area or overevaluated percentage of area covered by lakes. Nevertheless, in the present study, this site is removed only during the prediction of the reference variables and all sites are included in the rest of the analysis.

Figure 4 shows the fitting of the four reference variables by the PPR models. Cross-validation has selected PPR models with a unique direction $\alpha$ for all reference variables. The PPR equations that describe the relation between the reference variables and the site characteristics are explicit, for instance, the regression equation for the LCV has the form:

$$\log(\text{LCV}) = -1.80 + 0.26 \times f\left[-0.67 \times \log(\text{BV}) - 0.09 \times \sqrt{PLAC} \right.$$
$$\left. + 1.27 \times \log(\text{PLMA}) + 0.06 \times \text{LON} - 1.32\right] \tag{17}$$

Notice the constant term -1.32 and the norm of direction $|\alpha| \neq 1$ inside the function $f$ in Eq. (17). The difference between Eq (17) and the general form of the PPR model in Eq. (9) is the consequence of transformations on the explanatory variables. Indeed, during the optimization procedure of a PPR model, it is suggested to scale the explanatory variables in order to avoid the scale effect in the coefficients of the direction $\alpha$ (Hastie et al., 2009). Nevertheless, it is important to notice that the formula inside the function $f$ corresponds to a linear model.

Figure 4a shows a strong linear relationship between L1 and the predictor $\alpha'\mathbf{X}$. Conversely, Figures 4b,c,d show mild nonlinearity and hence indicate the need for more flexible models, such as PPR. The predictive performances of the reference variables are evaluated by the NHS criterion with values 91.5%, 33.3%, 6.7% and 55.7% respectively for L1, LCV, LSK and LKT. These results show that L1 is accurately predicted by the site characteristics, while a poor fit is associated to LSK. Indeed, Figure 4c suggests that apart from a few sites on the right of the curve, LSK appears not highly related to the predictor $\alpha'\mathbf{X}$. In comparison, linear models applied on the same reference variables lead to the following values of the NHS criterion: 90.9%, 28.2%, 7.8% and 48.1% respectively. Remark that the NHS criterion is calculated by

cross-validation, consequently even though the improved performances by the PPR method appear moderate they represent true fitting improvements.

Due to its poor fit, LSK may not be a proper reference variable for the delineation step. To validate this assumption, the neighborhoods are formed with and without using LSK and the rest of the analysis is carried out for both scenarios. Based on the RRMSE criterion, LSK must be maintained as it is associated to better predictive performances. This strategy is part of the backward stepwise selection procedure as described in section 3.1. Overall, it leads to discarding LKT and to maintaining L1, LCV and LSK. The second group of reference variables contains both the L-moments and the site characteristics. As with the first group, backward stepwise selection is performed and the final reference variables are: BV, PLAC, LCV and LSK. In order to distinguish the two groups of reference variables, RVN-LM will designate the first group with the L-moments only and RVN-HYB will designate the second group with both the L-moments and the site characteristics.

## 4.3 Results of the index-flood model

At this point, the steps 1-4 of the RVN methodology are performed and the neighborhoods are identified. Notice that for the RVN-LM method, the reference variables include the first three L-moments, which could be used as a moment estimator to deduce the target distribution. This approach is, however, not generally applicable to the present methodology as the reference variables are selected by a stepwise procedure. Moreover, it is necessary to identify a proper family of distributions from regional information, which is achieved here by analyzing the distribution of the gauged sites inside the neighborhoods. The index-flood model and the L-moments algorithm were proven to lead to a reliable procedure to identify a regional distribution and to estimate its parameters (Hosking and Wallis, 1997). In this model, the regional quantile corresponding to a return period $r$ at a target location $i$ is written $Q_i(r) = \mu_i Q(r)$, where $\mu_i$ is the index-flood. In the present study, the index-flood is taken to be the means of the at-site distributions and is predicted at the target location by multiple regression.

The index-flood model is fitted inside the neighborhoods obtained by each one of the four methods: ROI, CCA, RVN-LM and RVN-HYB. For CCA, two canonical pairs are calculated using flood quantiles corresponding to the 10- and 100-year return periods as hydrological variables, as described in section 2.1. The choice of the regional distribution is made between the four common families of distributions that were mentioned earlier: GEV, GLO, LN3 and P3. The parameters of the regional quantile function $Q(r)$ are calculated from the regional LCV and the regional LSK as the respective averages (see Eq. (12)). Figure 5a shows the L-moment ratio diagram for the regional LSK and LKT with RVN-LM. For each neighborhood, the distribution family is selected as the one having the nearest regional LKT to the theoretical value, given the regional LSK. RVN-HYB is omitted in Figure 5 to improve the clarity of the illustration, but has similar behaviour to RVN_LM.

Figures 5b,c,d present the L-moment ratio diagrams of the at-site LCV and LSK for three given target locations as an illustration of the gauged sites found in the respective neighborhoods. In these diagrams, the nearest gauged sites selected for RVN-LM, CCA and ROI are highlighted. Figure 5b shows that RVN_LM has a denser cluster of gauged sites in terms of LCV and is approximately centered on the true target. Conversely, Figures 5c and 5d show situations where the true

targets do not correspond to the predicted target. Although, all the reference variables are known at the target location for the ROI method, Figures 5b and 5c show that the selected sites are not located around the true target. This finding is coherent with the results of (GREHYS, 1996a, 1996b) which indicates that delineation according to physiographical similarity can lead to substantially different regions than delineation according to hydrological similarity.

Results of the cross-validation are presented in Figure 6. The evaluation criteria are calculated for every neighborhood with a size superior to 15 in order to calibrate the model. The tendency illustrated in this figure helps to visualize the evolution of these criteria with better perspective. The comparison of Figures 6a and 6b indicates that the optimal neighborhood sizes for RRMSE and NHS are not always in agreement. In particular, the best RRMSE for the RVN-HYB method is with 24 sites, while the best NHS is with nearly 80 sites. Nevertheless, the optimal values for the three other

methods are obtained with approximately 30 sites for both criteria. Figure 6b indicates that all methods have a relatively stable NHS between 86% and 87%, but the best NHS is obtained by RVN-LM. Conversely, Figure 6a shows clearer improvements of the calibration in terms of the RRMSE criterion. Hence, the calibrated models are set according to the RRMSE criterion and are represented by circles in Figure 6. The results are summarized in Table 1. RVN-HYB, with a RRMSE of 40.1% outperforms the other methods. In particular, a difference of 6.1% and 5.3% is observed respectively with

the traditional ROI and CCA methods.

        Figures 6c,d present respectively the AHM and the ARE criteria obtained from the considered methods. The AHM criterion indicates that the ROI and the CCA methods have in general lower heterogeneity than the whole dataset, but are largely outperformed by the RVN-LM and RVN-HYB methods especially for smaller neighborhoods. This is not surprising as the RVN-LM and RVN-HYB pool together sites with similar L-moments, but this quantifies the intuitive assumption that

the regional LCV is calculated with less uncertainty when the L-moments are directly considered instead of other reference variables. In particular, the AHM of the ROI method is 72.8% with the optimal neighborhood size of 30. In comparison, the AHM of the RVN-LM method is 14.5% with the optimal neighborhood size of 29 sites, which is considerably lower. Figure 6c shows that the AHM criterion of the RVM-LM method does not reach a similar level to the ROI method until as much as 120 sites are used. These results indicate that even for relatively small neighborhoods, the ROI method identifies regions

that are only slightly less hydrologically heterogeneous than all sites pooled together. This suggests that, in the present case study, the ROI method has difficulties identifying sites that are similar to the target site in terms of LCV.

        As mentioned in section 4.2, previous studies have identified a few problematic stations in the considered dataset. Figure 7 presents the residuals between different methods. As it may be difficult to see small improvements by uniquely observing points around the $y = x$ lines, the visualization of Figure 7 is improved by adding a flexible fit of the point

cloud, using a standard smoothing spline approach. The resulting red lines indicate, if close to $x$, the residuals are lower in average for one of the two methods. In general, the points associated to the largest relative discrepancies are close to the $y = x$ line, which indicates that the sites that are difficult to predict are essentially the same for all methods. However, Figures 7a,b show that the RVN-HYB method specifically improves the prediction of the sites with the lowest and largest relative discrepancies as the red line is clearly located under the $y = x$ lines, which explains the improved RRMSE in

Table 1. On the other hand, Figures 7c,d indicates that at the logarithmic scale, the RVN-LM method achieved predicted values that are mostly similar to the ROI and CCA methods, which explains the similarity of the NHS criteria for all the compared methods.

        The present case study is an example of a region where some sites are problematic for any method. In practice, the

residuals are not known, consequently we do not know if the target sites of interest will be "problematic" or not. Globally, what Figure 7a indicates is that the RVN-HYB model is somehow more robust, because for the sites that are well predicted by simpler models, such as ROI, RVN-HYB will perform in average similarly. However, if the target site is predicted less accurately, the RVN-HYB model will (in average) be better in terms of RRMSE. Consequently, the overall gain may seem of moderate magnitude. However, for some problematic stations the gain could be more substantial. In particular, the red lines in the left part of Figure 7a appear mostly influenced by two points, but the two improvements are of 77.2% and 68.5%, which is considerable.

## 4.4 Results of the regression-based model

Prediction of Q100 at the target location is also carried out by the regression-based model using the same delineation methods as with the index-flood model, but with potentially different calibration values for the neighborhood sizes. Consequently, the description of steps 1-4 (in section 3.1) are identical to those of the index-flood approach and are not repeated here.

Cross-validation criteria for the regression-based model are presented in Figures 8 and summarized in Table 1. As with the index-flood model, Table 1 reveals that the RVN-HYB method leads to the best performance in terms of the RRMSE. Although all methods differ by less than 2% in terms of NHS, results indicate that NHS values corresponding to CCA and RVN-HYB are inferior to those corresponding to the regression model applied on all gauged sites, which corresponds to $n = 150$ in Figure 8b. However, CCA leads to the best relative efficiency as indicated by the ARE criterion in Table 1. Hence, CCA corresponds to the regression models with, on average, the lowest uncertainties. This indicates that flood quantiles may be better reference variables for the regression-based model than for the index-flood model and suggests that in general different reference variables may be more appropriate for different situations. Nevertheless, the two close lines in Figure 8d reveal that, for the same neighborhood size, the RVN-LM method has ARE values that are similar to CCA. In terms of AHM, Figure 8c is identical to Figure 5c except that new neighborhood sizes are indicated in circles.

The fit of the regression-based model is graphically assessed in Figure 9 by Quantile-Quantile plots. It is shown that, for all the delineation approaches, the regression-based models correctly predict the flood quantile Q100 at target locations as it correctly follows the $y = x$ line. However, the comparison between the methods ROI and RVN-HYB shown in Figures 8a,c and the methods CCA and RVN-LM shown in Figures 8b,d do not show clear differences. A more precise assessment would be obtained by comparing the residuals instead, as it is done in Figure 7. However, the predictions of the regression-based models are very similar to those of the index-flood models and they lead to very similar figures, which are not reported here. Table 1 provides also a comparison of the performances of the index-flood and the regression-based models. In terms of RRMSE and NHS criterion, the two approaches lead to very similar results, which is coherent with what it is reported in other studies (GREHYS, 1996a, 1996b; Haddad and Rahman, 2012). Therefore, similar conclusions can be drawn from the two approaches. For instance, in both cases, the RVN-HYB method leads to the best results in terms of RRMSE.

## 5. Conclusions

A general methodology was investigated to improve homogenous properties of neighborhoods in RFFA. A procedure to calculate relevant reference variables at a target location prior to the RFFA was proposed to improve neighborhood properties and to reduce uncertainties. The predicted values of reference variables represent the unknown centers of neighborhoods delineated according to a distance of gauged sites with respect to the centers. The proposed method represents a generalization of both ROI and CCA methods in RFFA. The proposed RVN method has the advantages of accepting various groups of reference variables, of considering nonlinear interrelations and of being more objective since L-moments are used instead of estimated flood quantiles from at-site analysis.

In this study, the reference variables correspond to transformed L-moments. The resulting RVN-LM and RVN-HYB methods were applied on sites located in the southern part of the Province of Quebec, Canada, to predict flood quantiles corresponding to the 100-year return period by both index-flood and regression-based models. The prediction of the reference variables at target locations showed that, after proper transformations, L1 can be linearly related to the site characteristics, but no proper transformations are found for the other L-moments. This justifies the consideration of the PPR method to account for the nonlinearity in the prediction of the reference variables. In general, other models, such as generalized additive models or artificial neural networks, could be considered instead of PPR to account for the nonlinearity. Nevertheless, the PPR approach unveils direction vectors that provide explicit, parsimonious and meaningful regression equations.

Although none of the methods performed best for all criteria, cross-validation showed that the proposed RVN method performs well in comparison to the traditional ROI and CCA methods. In both the index-flood and the regression-based model the best RRMSE is obtained by RVN_HYB and the best NHS is obtained by RVN_LM. In particular, the favorable RRMSE values obtained by RVN-HYB are due to a more robust estimation of problematic sites. However, RVN_LM has the best balance, because it achieves the best or the second best values for all criteria. Most importantly, the utilization of hydrological reference variables with the CCA and RVN methods has reduced the uncertainty on the regional LCV, the index-flood and the predicted flood quantiles, in comparison to ROI. Consequently, prior modeling of hydrological reference variables was shown to be advantageous to the delineation of neighborhoods in RFFA.

The present study has made specific assumptions in order to investigate the RVN method in well-defined conditions. Nevertheless, the approach that consists in predicting hydrological reference variables in an *a priori* analysis remains valid when other choices of regression models, neighborhood forms and metrics are considered. More comparative studies should be carried out to evaluate alternatives to fixed size neighborhoods and Euclidian distances in the specific context of the RVN framework.

The L-coefficient of skewness is commonly used in RFFA to describe the shape of a distribution. Consequently, to improve the result of the RVN method, further research efforts could focus on improving the prediction of this crucial reference variable. One way to improve the prior analysis of the hydrological reference variables is the consideration of the unequal sampling error. This aspect is often considered in the estimation of flood quantiles in RFFA, but may also play an important role in the prior analysis of the RVN method.

**Acknowledgement**

Financial support for this study was graciously provided by the Natural Sciences and Engineering Research Council (NSERC) of Canada. The authors are graceful to the Editor, Dr. Elena Toth and the two reviewers whose comments and suggestions contributed to the improvement of the manuscript.

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

**Table 1: Evaluation criteria for the RVN method for optimal neighborhood sizes.**

| | Model | Size | RRMSE | NHS | AHM | ARE |
|---|---|---|---|---|---|---|
| **Index-Flood** | | | | | | |
| | ROI | 30 | 46.2 | 86.5 | 72.8 | 57.3 |
| | CCA | 28 | 45.4 | 86.2 | 41.7 | 42.9 |
| | RVN-LM | 29 | 45.0 | **87.1** | **14.5** | **36.9** |
| | RVN-HYB | 24 | **40.1** | 86.2 | 16.5 | 43.1 |
| **Regression-based** | | | | | | |
| | ROI | 30 | 44.9 | 86.9 | 72.8 | 64.7 |
| | CCA | 28 | 43.5 | 86.1 | 41.7 | **30.6** |
| | RVN-LM | 39 | 41.7 | **87.6** | 17.9 | 39.8 |
| | RVN-HYB | 24 | **39.5** | 86.2 | **16.5** | 42.5 |

Best criteria in bold

**List of Figures**

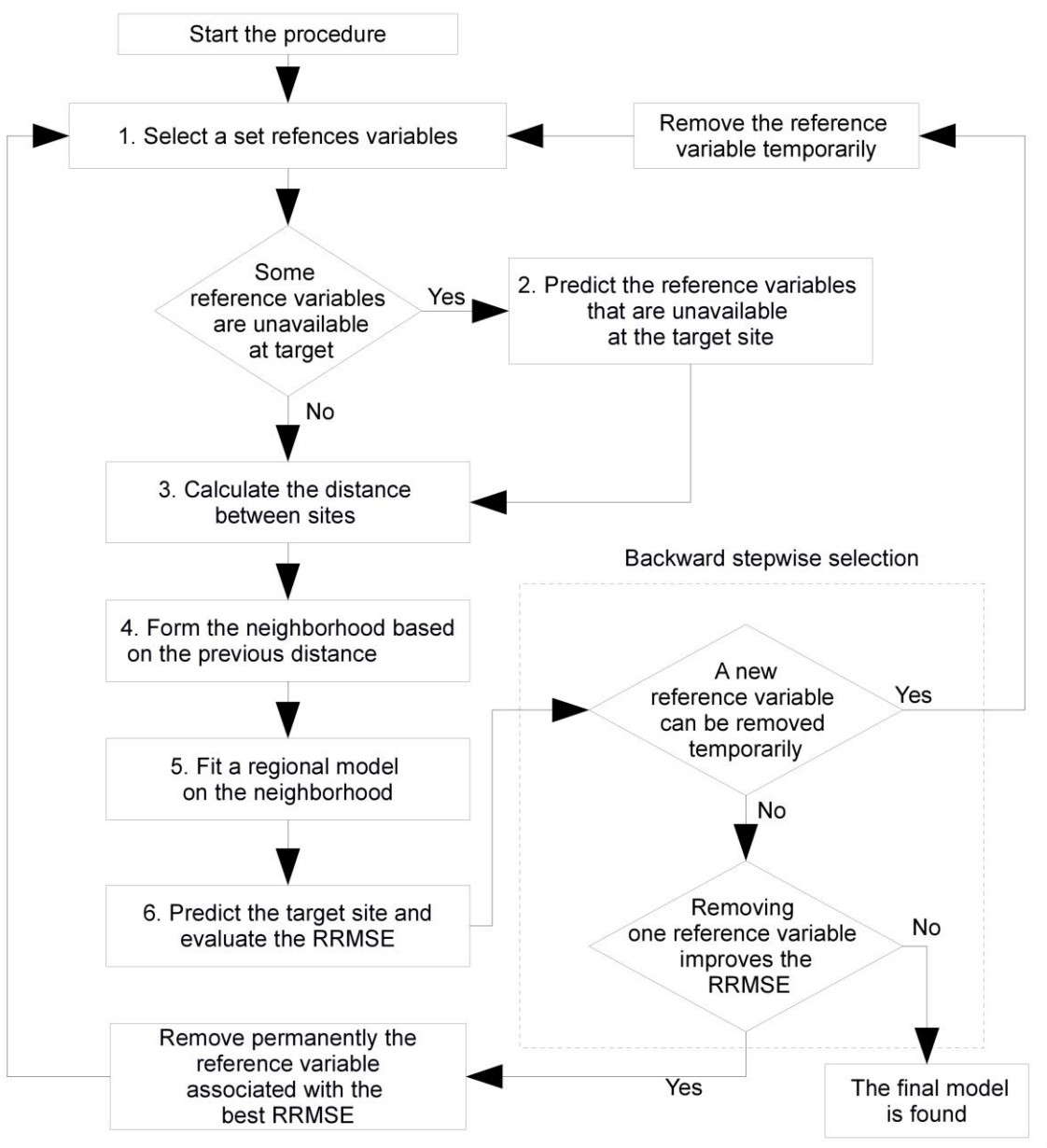

**Figure 1: Diagram of the RVN method using backward stepwise selection.**

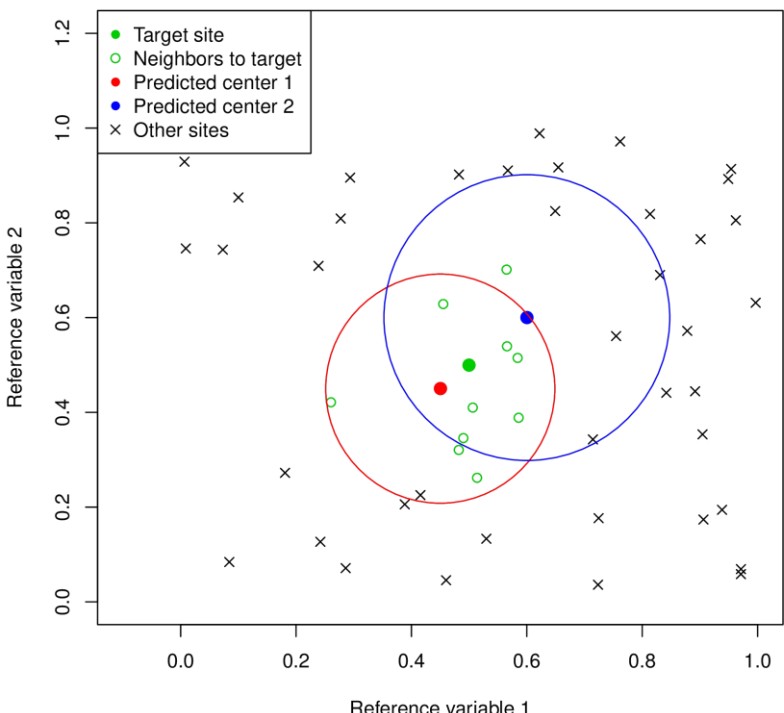

**Figure 2: Illustration of the neighborhoods obtained by the RVN method.**

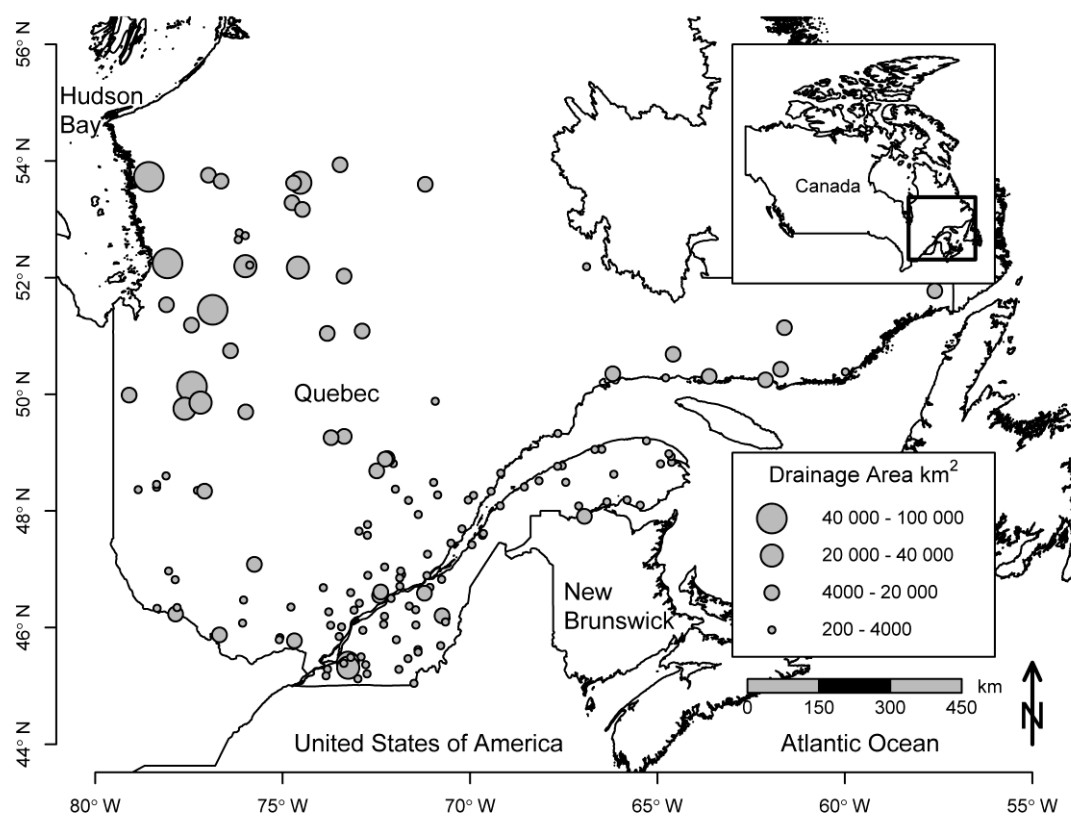

**Figure 3: Location of the 151 hydrometric stations in Southern Quebec, Canada.**

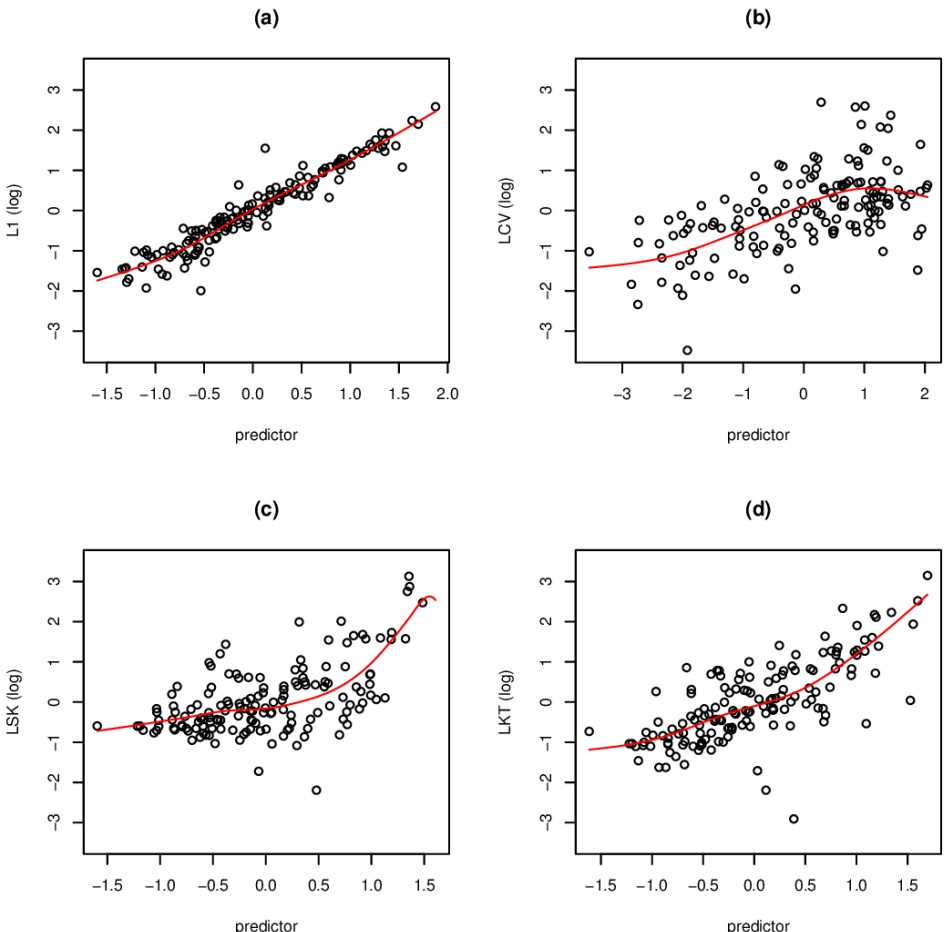

**Figure 4: Residuals of the reference variables by PPR methods.**

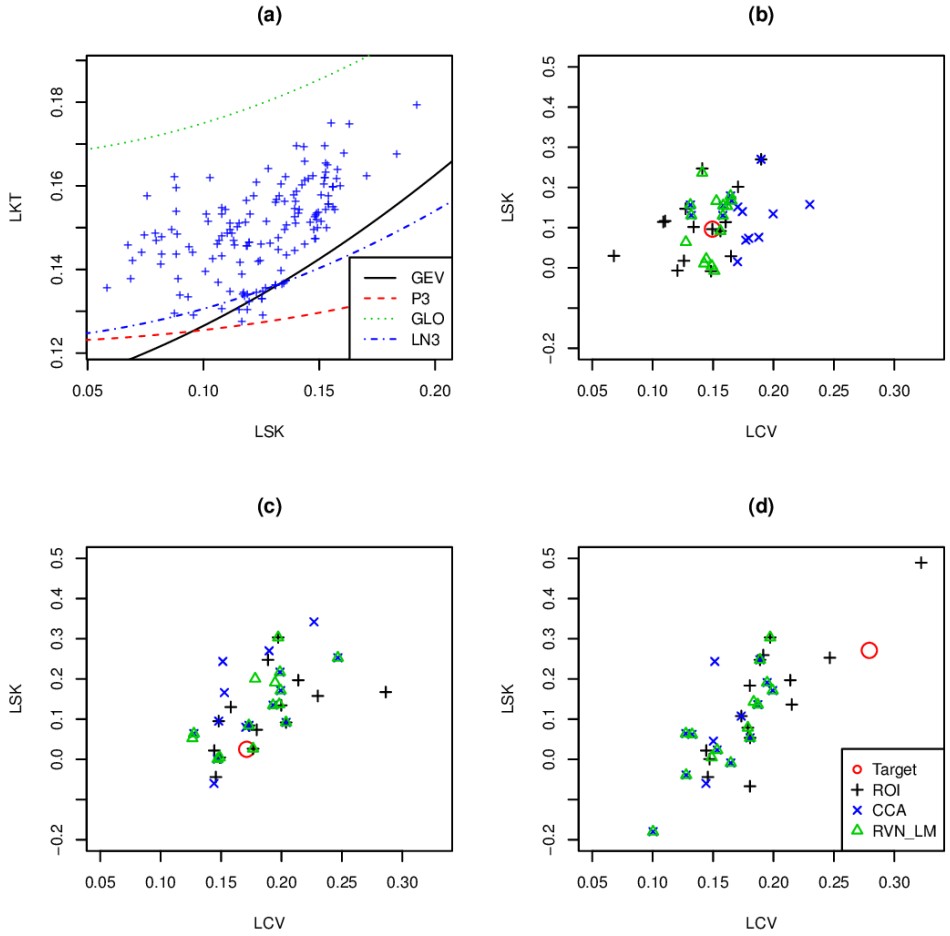

**Figure 5: L-moments ratio diagram for index-flood model. (a) Regional L-moments for RVN-LM with 29 gauged sites. (b),(c) and (d) Regional L-moments based on the 15 nearest gauged sites for 3 selected target locations.**

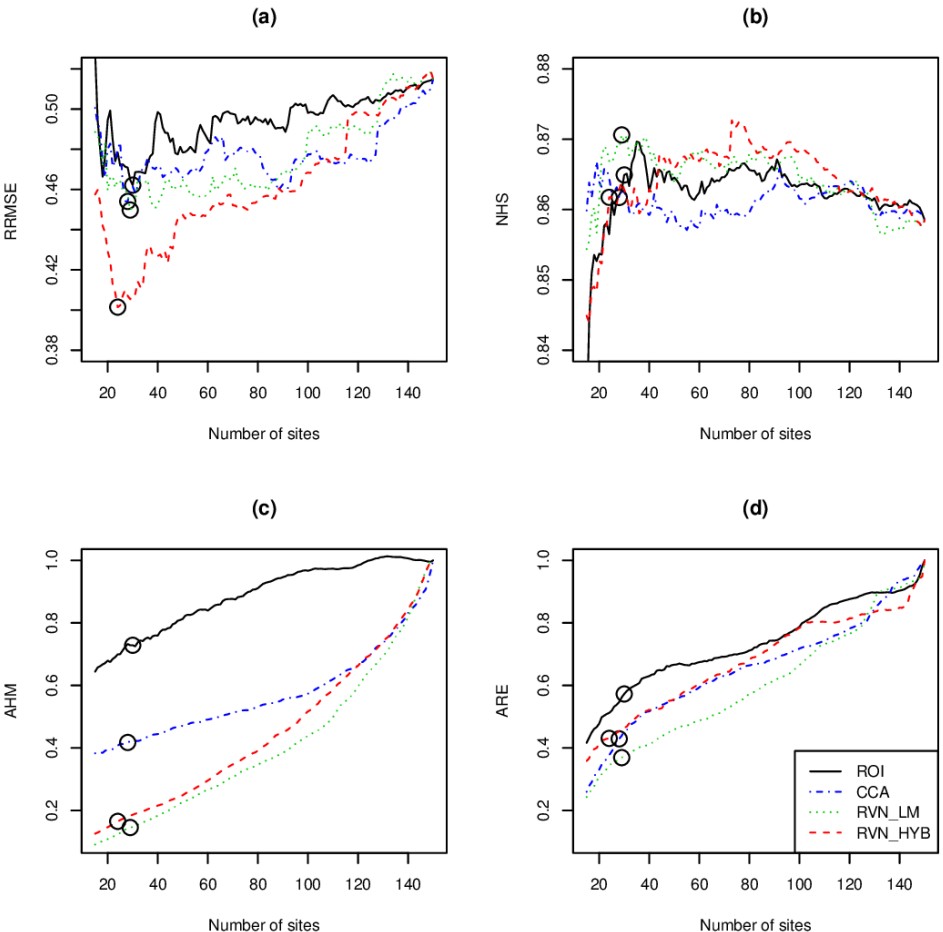

**Figure 6: Evaluation criteria for the index-flood model. Calibrated models are represented by circles.**

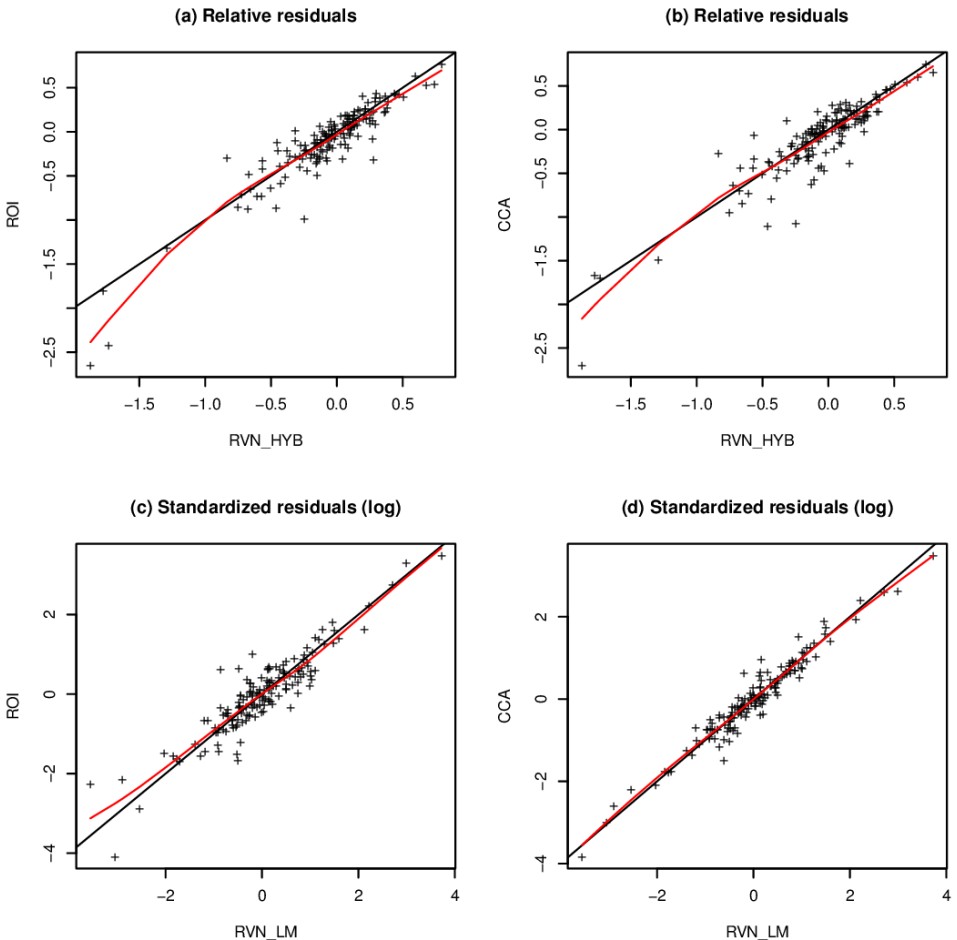

**Figure 7: Comparison of the cross-validation residuals for Q100 for different methods. The black line is the unitary slope and the red line is a smooth fitting of the residuals.**

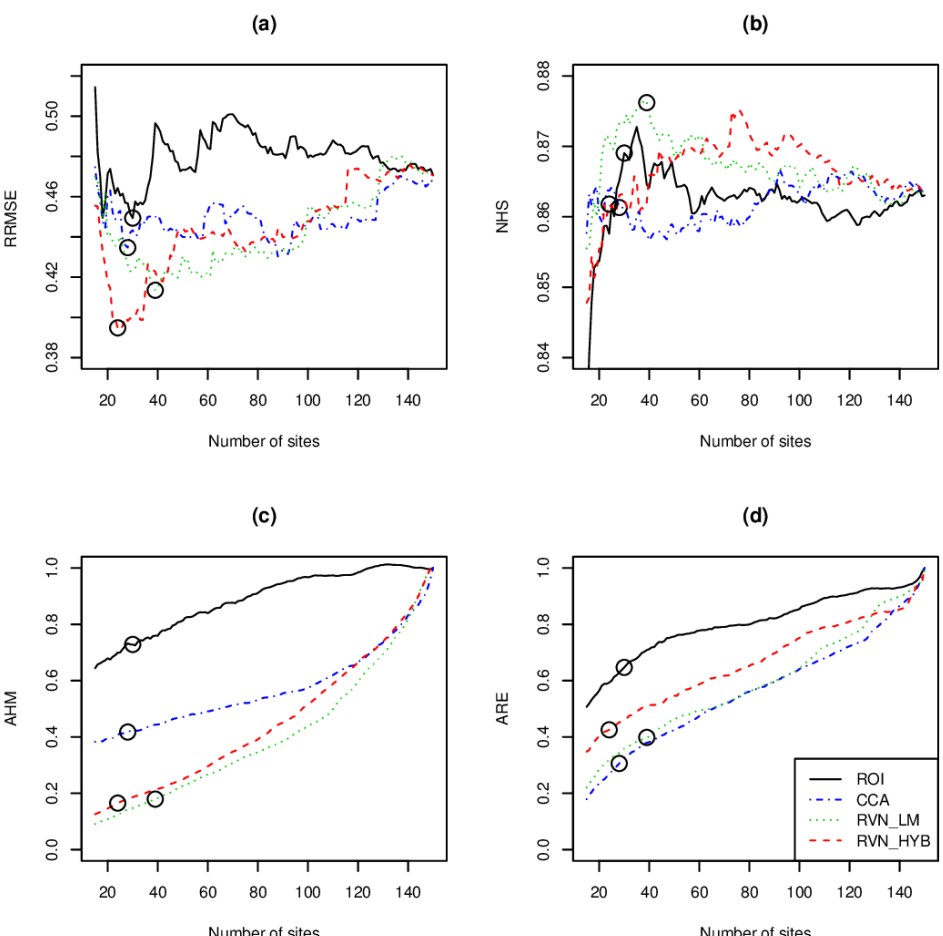

**Figure 8: Evaluation criteria for the regression-based model. Calibrated models are represented by circles.**

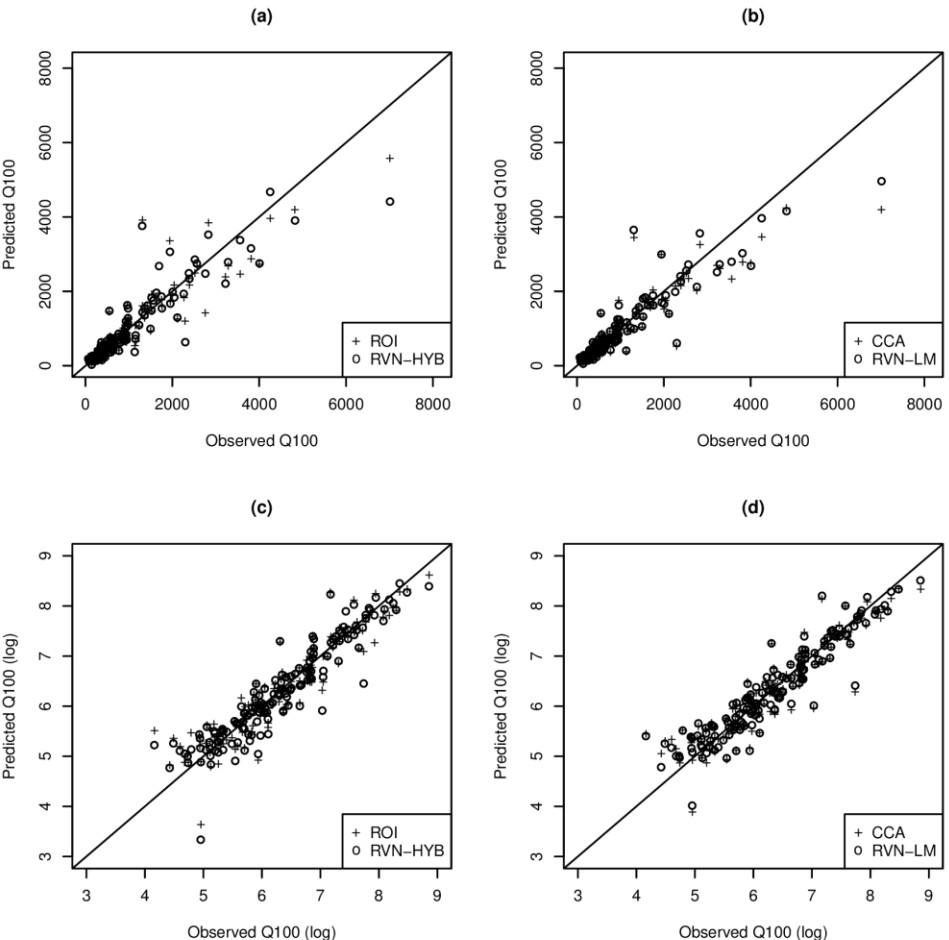

**Figure 9: Quantile-Quantile plot of Q100 for the RVN method with regression-based model.**