# Peer review of "Delineation of homogenous regions using hydrological variables predicted by projection pursuit regression"

_Hydrology and Earth System Sciences, 2016_

## Referee Comment (RC1) · Anonymous Referee #1 · 20 May 2016

General comments

The article "Delineation of homogeneous regions using hydrological variables predicted by projection pursuit regression" by Durocher et al. describes an improvement of existing techniques for the regional estimation of flood quantiles. The topic is very relevant, but I found the manuscript not completely clear in some parts and with some methodological flaws. While readability (see comment n. 1) can be improved with a revision of the text, some methodological issues would require a complete reanalysis of the work. My main concerns (see comments n.2 and 3) are basically related to the use of a very complex procedure that is not justified by the results. This finding (i.e., that the procedure does not really produce improvements) would be a result itself, but the authors

seem to overlook it to support their initial hypotheses. For these reasons, I suggest a rejection of manuscript.

Major comments

1. While the aim of the work is very clear, I found quite intricate the description of the operational procedure. The point list on page 6, and in particular the step i) (which is the main focus of the paper) should be supported by a quantitative example to make the procedure easier to understand. For instance, the plots in figure 3 could be used in this part of the manuscript to better describe how the procedure works (and not only from page 11 to comment results). Moreover, step i) seems a kind of "preliminary" regionalization of the L-moments of the target site. Such L-moments are then used to support the delineation of the region. Why such preliminary estimates cannot be directly used in the prediction of flood quantiles? This point should be discussed by the authors, highlighting the possible differences with the direct estimation of the quantiles based on preliminary L-moments.

2. The whole procedure is rather complex, so I would expect the proposed method clearly outperforming the others. However, looking at figure 5, it seems that the residuals are scattered more or less homogeneously around the bisector, meaning that the "traditional" and the "new" methods performs, in average, the same. Figure 5 tells me that there is no significant difference between the methods, so I would use the simpler one. Of course, by computing an error metric over the whole set of residuals one may obtain slightly better performances of the RVN models (but this is not reported; a summary table would be appreciated). The authors state on page 12, line 25 onwards, that improvement is effective for sites with largest discrepancies. This seems not true except for two point in figure 5a and one point in figure 5b (all the points in the bottom-left corner of each panel). Figures 5c and 5d show points equally distributed around the bisector also in the bottom-left corner. Hence, is the complexity of the RVN model justified by a so small performance improvement?

3. On page 10, line 29, the nonlinear relationship between the (transformed) predictors and the (transformed) L-moment is mentioned and the authors say that it is shown in figure 2. This non-linear relationship would justify the use of a spline interpolator, but actually this is a questionable point. In fact, figure 2 tells a different story. Panel a clearly show a linear relationship (in the transformed variables; this is expected as often the mean value can be linearized with log transformations). In the b, c and d panels there is a much larger scattering, which does not allow to identify a clear complex pattern, even if all the plots show an increasing trend. Looking at the scatter plots I believe that most of the people would adopt a simple linear regression (said with 2 parameters) which is much more stable and robust. My personally idea is that the choice of the authors is not justified and that a linear model should be at least compared to the spline interpolator.

Minor comments

P7 L6 Please, give a more detailed description of "true neighborhood" meaning.

P11 L5-7 I found quite strange that the L-kurtosis performs much better that the L-skewness as in general the prediction ability deteriorates with increasing order of L-moments. The authors should investigate in more detail this issue.

Sections 2.1 and 2.2 are rather obvious but useful for the paper, so they could be placed in appendix and referred to in the main text. I would also add at the end of the "Multiple regression" section a note about weighted and generalized least squares.

Figure 5. Not clear which kind of information is provided by the "smooth fitting of the residuals". Also in this case, the smooth fitting seems too complex tool which does not add any further information.

---

## Referee Comment (RC2) · T. Gado (Referee) · 22 May 2016

General Comments:

The present manuscript investigates the utilization of hydrological information in Regional Frequency Analysis (RFA) in order to improve homogenous properties of neighborhoods and then improve regional flood estimation.

In general, the paper is well organized and the contribution of the study is relevant. Nevertheless, throughout the manuscript, the authors support the idea of using the estimation of hydrological variables, instead of site characteristics, to delineate homogeneous regions. Yet, the estimation of hydrological variables is based on subjective selections of site characteristics and subject to model errors. Thus, using such estimations to improve homogenous properties of neighborhoods in RFA is questionable. Consequently, the improvements in the results are insignificant as mentioned by the referee #1. Moreover, the manuscript is not clear in some parts especially in the methodology and the results. In my opinion, the article may need major revisions before publication. Major and specific comments are shown below.

Major Comments:

1. The literature is not complete and does not state what other researchers have done in order to improve the flood estimation at ungauged sites. So, the authors should improve the description of the existing literature on the topic investigated. In particular, the manuscript should elaborate a little bit better on the evolution of the ROI method as the study focuses on the neighborhood approach for homogenous region delineation.

2. The methodology is blurred, difficult to follow, and contains some odd judgements. For instance, LSK was maintained because it is associated to better predictive performance, however, it is poorly predicted by the site characteristics (P11 L5 – 12). Also, the authors did not show some details such as the additional translation which necessary to avoid numerical difficulties of LSK and LKT due to negative values (P10 L19).

3. Although the authors introduced a complicated methodology, they did not make enough efforts to clarify the description of the results; such as confusing explanation of Fig. 4 (P12 L8 – 17) (e.g., why 80 sites? in P12 L12), and unclear Fig. 5 and its explanation (P12 L26 – 33). Furthermore, the presentation of the results of the regression-based model needs improvements to be clearer (P12 L35 – P13 L10). I recommend using the simple Q-Q plot to assess the compared methods regarding the estimation of regional flood quantile. Also, the results should contain numerical tables to quantitatively clarify the differences between the considered methods. The authors can find a close example for the presentation of such results in the reference Gado and Nguyen (2016). Finally, comparing the results of the index flood and the regression

methods would be valuable here.

Specific comments:

1. P1 L16 – 17. Which properties does the hydrological information in Regional Frequency Analysis enforce for a group of gauged stations? I suggest to add "desired properties".

2. P1 L18. Ungauged sites can be defined by site characteristics in the neighborhood delineation methods (e.g., ROI). Therefore, there is no a challenge for using neighborhoods in RFA regarding the unavailable hydrological information at ungauged sites.

3. P1 L23. The regional frequency analysis can be applied for flood or extreme rainfall or any other extreme events. Hence, it should be stated that the case study is for regional flood estimation.

4. P2 L21. "the distance between hydrological variables". The distance is between locations not variables.

5. P3 L4. "as an estimation model"

6. P9 L4. Please, define NH in equation 14.

7. P9 L14 – 15. How can a regression model fitted on two different neighborhoods, for the same target location, obtain identical values?

8. P10 L2. I don't believe that 15 years of data are enough to get statistically reliable results, why did authors choose 15 years as the minimum time series used in the study.

9. P10 L3. I think you should have at least a map showing the locations of the selected stations in the case study (Quebec).

10. P10 L7. Using the maximum likelihood for parameter estimation with small time series (e.g., 15 years) may cause convergence problems, I would recommend using L-moments instead.

11. P11 L16. What does HYB denote for in "RVN-HYB"?

12. P11 L19. "One of the objectives of RFA is to identify a proper family of distributions from regional information" This is not an objective of the RFA. I suggest to write one of the main steps.

13. P11 L23. Please, define here the Q(r) as the regional quantile.

References:

Gado, T. A., and Nguyen, V.T.V., 2016. Comparison of homogenous region delineation approaches for regional flood frequency analysis at ungauged sites. J. of Hydrol. Eng., 21(3), Doi: 10.1061/(ASCE)HE.1943-5584.0001312.

---

## Editor Comment (EC1) · E. Toth (Editor) · 27 May 2016

Both referees (very expert researchers in the field of the submitted work) find that the proposed procedure, in addition to being not clear in its description, is not justified by the results and also methodologically questionable. A comparison with more consolidated and simple approaches, along with an improvement in the description, would certainly be needed in case the paper were to be resubmitted, but the main concerns of the referees need to be addressed thoroughly and urgently, since they are both very critical on the work and suggest either rejection or very major revision work. I would suggest to the Authors to reply in a few days to the Referees' comments, so that the Referees may, if they have the time and opportunity, reply back. I will ask the Editorial

Office for an extension of the deadline, to give the Authers and the Referees more time for discussing such critical points. Elena Toth HESS Editor

---

## Author Comment (AC1) · 1 Jun 2016

** see the attached PDF**

Major (1.1) Reviewer: "The point list on page 6, and in particular the step i) (which is the main focus of the paper) should be supported by a quantitative example to make the procedure easier to understand. For instance, the plots in figure 3 could be used in this part of the manuscript to better describe how the procedure works (and not only from page 11 to comment results)." The authors agree that the illustration provided by Figure 3 is useful to understand this step of the methodology. The Figure 1 was initially intended to provide a schematic illustration of this step (i). Figure 1 will be improved in a revised version of the manuscript in order to be more similar to Figure 3 in its

interpretation. (1.2) Reviewer: "Moreover, step i) seems a kind of "preliminary" regionalization of the L-moments of the target site. Such L-moments are then used to support the delineation of the region. Why such preliminary estimates cannot be directly used in the prediction of flood quantiles? This point should be discussed by the authors, highlighting the possible differences with the direct estimation of the quantiles based on preliminary L-moments." In a situation where the target distribution is assumed to be known (i.e. the same family of distribution is assumed everywhere) the preliminary prediction could be used indeed to deduce the parameter of the target distribution from the L-moments. However, in the framework the distribution is unknown and the L-moments cannot be used directly to estimate the flood quantile without specifying a family of distributions, which is achieved here by using the information of relevant neighborhoods. The comment of the reviewer also applied only to the utilization of the index-flood model inside the neighborhood. In a revised version of the manuscript, the authors agree to add a comparison between quantiles estimated by the parameter of the PPR method and the regional estimates. (2.1) Reviewer: "Figure 5 tells me that there is no significant difference between the methods, so I would use the simpler one. Of course, by computing an error metric over the whole set of residuals one may obtain slightly better performances of the RVN models (but this is not reported; a summary table would be appreciated). "

Notice that two error metrics are actually used in the present study: RRMSE and NHS. The authors agree that an additional table will help to summarize more directly the information provided in the text and in Figure 4 and 6 for the calibrated model. The Table below will be included and discussed in a revised version of the manuscript.

Table 1:

| Model | RRMSE | NHS | AHM | ARE |
|---|---|---|---|---|
| Index-Flood ROI | 46.2 | 86.5 | 72.8 | 57.3 |
| CCA | 45.4 | 86.2 | 41.7 | 42.9 |
| RVN-LM | 45.0 | 87.1 | 14.5 | 36.9 |
| RVN-HYB | 40.1 | 86.2 | 16.5 | 43.1 |
| Regression-based ROI | 44.9 | 86.9 | 72.8 | 64.7 |
| CCA | 45.2 | 86.4 | 40.7 | 26.7 |
| RVN-LM | 42.6 | 87.3 | 14.2 | 34.3 |
| RVN-HYB | 42.3 | 86.2 | 21.2 | 51.1 |

Best criteria in bold

(2.2) Reviewer: "The authors state on page 12, line 25 onwards, that improvement is

effective for sites with largest discrepancies. This seems not true except for two point in figure 5a and one point in figure 5b (all the points in the bottom-left corner of each panel). Figures 5c and 5d show points equally distributed around the bisector also in the bottom-left corner. Hence, is the complexity of the RVN model justified by a so small performance improvement?"

The authors want to make the precision that the comment "improvement is effective for sites with largest discrepancies" applied only to the relative residuals as this comment is made while describing Figure 5a and 5b. To make it clearer, the terms "largest relative discrepancies" will be used in a revised manuscript. The additional complexity of the RVN method in respect of the traditional ROI method concern only the preliminary step (i), where the missing hydrological information is substituted by predicted values. The question is thus if the addition of the preliminary step is justified. Notice that RRMSE and NHS are cross-validated criteria and hence they are criteria that Ânpenalized̂ excessive complexity in the prediction. Consequently a better RRMSE implies that a model truly brings additional information on the quantiles. The overall improvement in terms of RRMSE for the RVN-HYB method in respect of the ROI and the CCA method is respectively 6.1% and 5.3% (see table in major comment 2.1 or Figure 4). Although of moderate amplitude, the authors believe that these improvements are nonetheless substantial. In particular, the authors agree with the reviewer that the red line in the bottom-left corner of Figure 5a and 5b are mostly influenced by few points. Nevertheless, in Figure 5a, the 4 largest relative discrepancies are better predicted by RVN-HYB and the two best relative improvement are of 77.2% and 68.5%. In Figure 5, small changes in average can be difficult to assess by the naked eye. The red lines are smooth curves fitted between the residuals of the two models is there to indicate which models in average have locally lower residuals. It can be seen that for the residuals above approximately 0.2 ( i.e. 20% of the observed values) the red line is distinctively under the bisector (upper right corner) and this is true for several points. This means that the sites that are overestimated are in average less overestimated by the RVN-HYB method. Actually, 8 out of the 9 residuals located at the most right are

better predicted by the RVN-HYB model. Similar behavior is also noticeable in upper-left corner of Figure 5b. The present case study is an example of a region where some sites are problematic for likely any models. In practice, the residuals are not known, consequently we don't know if the target sites of interest will be "problematic" or not. Globally what the results and Figure 5a indicate is that, in a certain way, the RVN-HYB model is more robust, because for the sites that are well predicted by most models, including ROI, RVN-HYB will perform in average similarly. However if the target site is predicted less accurately, the RVN-HYB model will, in average, be better in terms of relative errors. Consequently, the overall gain may seem of moderate in magnitude, but for some problematic station the gain is more substantial. In a revised version of the manuscript the meaning of Figure 5 and the red line will be better explained and the arguments above will be added.

(3) Reviewer: "On page 10, line 29, the nonlinear relationship between the (trans-formed) predictors and the (transformed) L-moment is mentioned and the authors say that it is shown in figure 2. This non-linear relationship would justify the use of a spline interpolator, but actually this is a questionable point. In fact, figure 2 tells a different story. Panel a clearly show a linear relationship (in the transformed variables; this is expected as often the mean value can be linearized with log transformations). In the b, c and d panels there is a much larger scattering, which does not allow to identify a clear complex pattern, even if all the plots show an increasing trend. Looking at the scatter plots I believe that most of the people would adopt a simple linear regres-sion (said with 2 parameters) which is much more stable and robust. My personally idea is that the choice of the authors is not justified and that a linear model should be at least compared to the spline interpolator." The authors understand the concern of the reviewer, but want to add the remark that the CCA model is actually the model that linearly predict the reference variables. Therefore, in the case of the index-flood model, the impact of assuming nonlinearity is measured by comparing the CCA and the RVN-LM model. These points could be made clearer in the revised version of the manuscript. Nevertheless the authors understand the concern of the reviewer in the

used of the PPR in Figure 2. In that case, the NSH for the two regressions method are provided in the table below. Table 2: NHS(%) L1(log) LCV(log) LSK(log) LKT(log) PPR 91.5 33.3 6.7 55.7 linear 90.9 28.2 7.8 48.1

For L1 and LSK the difference between the NHS criteria is about 1 %, which the authors agree is very similar. However, the gains for the LCV is of 5.1% and of 7.6% for the LKT are substantial. Notice that the NHS is a cross-validation criteria and hence this result does not represent a form of overfitting in favor of the more complex PPR approach as the prediction are obtained without the use of the predicted sites. Unnecessary complexity is then ÂńpenalizedÂż by such criteria. The fitting of LCV and LKT are cases of mild nonlinearity, but such mild nonlinearity can be adjusted here because 151 is a reasonable number of sites. The authors agree that the situation would be different if for instance only 20 sites where available. Therefore, in the present case study, it is false to say that linear models are more stable and robust than PPR as the performance is assessed by cross-validation. In a revised version of the manuscript the term "nonlinearity" will be changed for "mild nonlinearity" to be more precise. The interpretation of the criteria NHS as cross-validation criteria will be underlined and the particularity of having a sufficient number of sites to use PPR will be made clearer. Minor (1) Reviewer: "P11 L5-7 I found quite strange that the L-kurtosis performs much better that the L-skewness as in general the prediction ability deteriorates with increasing order of L-moments. The authors should investigate in more detail this issue." The authors, agree with the reviewer that L-skewness is expected to be, in general, better predicted than the L-kurtosis. The data have been investigated for data manipulation errors and nothing have been found. It appears to be a legitimate exception to the rule. (2) Reviewer: "P7 L6 Please, give a more detailed description of "true neighborhood" meaning." To clarify the term "true neighborhood" the following lines would be added to a revised manuscript around P7-L5: "If the hydrological variables were known at the target location, the distance would be available and the neighborhood that truly regroups the most hydrologically similar sites to the target location could be formed. However in practice this true neighborhood is unknown. Using instead an estimate

has the effect of falsely including that are less hydrologically similar than the sites that would be included in the true neighborhood." (3) Figure 5. Not clear which kind of information is provided by the "smooth fitting of the residuals". Also in this case, the smooth fitting seems too complex tool which does not add any further information. As briefly discussed in the major comment (2.2), at the proximity of a point say (X,X) in Figure 5, several points maybe under or above the bisector, which makes seeing small improvement for a method difficult. The smooth fitting provided by the red lines indicates specifically the average between the residuals of two models in the proximity of a point X. The authors agree with the reviewer that parsimony is important. The smoothing splines procedure used here as a visual guide is widely available in most numerical softwares and the "degree of complexity" is controlled by the generalized cross-validation criteria (GCV), which is a widely used criteria to avoid overfitting. Hence, with the information from 151 sites, the authors believe that the smooth fitting is not "too complex", but "as complex" as the residuals suggest it. More details on the interpretation of the red lines will be provided in a revised version of the manuscript.

Please also note the supplement to this comment:
http://www.hydrol-earth-syst-sci-discuss.net/hess-2016-123/hess-2016-123-AC1-supplement.pdf
* * *

---

## Author Comment (AC2) · 1 Jun 2016

** see attached pdf**

Major

(1) The literature is not complete and does not state what other researchers have done in order to improve the flood estimation at ungauged sites. So, the authors should improve the description of the existing literature on the topic investigated. In particular, the manuscript should elaborate a little bit better on the evolution of the ROI method as the study focuses on the neighborhood approach for homogenous region delineation. The authors agree with the reviewer and a revised version of the manuscript will in-

clude an improved description of the existing literature on ROI. Notice, however, that much effort in the recent literature of ROI method concern the problem of estimating the model by generalized least squares, which is not the problem addressed in the present study. See for instance Reis, D., Stedinger, J., Martins, E., 2005. Bayesian generalized least squares regression with application to log Pearson type 3 regional skew estimation. Water Resources Research 41. Griffis, V., Stedinger, J., 2007. The use of GLS regression in regional hydrologic analyses. Journal of Hydrology 344, 82–95. Kjeldsen, T.R., Jones, D.A., 2009. An exploratory analysis of error components in hydrological regression modeling. Water Resources Research 45, n/a–n/a. doi:10.1029/2007WR006283 Moreover, the main focus is more the improvement of the ideas behind the CCA method, for which the recent developments are included in the introduction.

(2.1) The methodology is blurred, difficult to follow, and contains some odd judgements. For instance, LSK was maintained because it is associated to better predictive performance, however, it is poorly predicted by the site characteristics (P11 L5 – 12). The authors believe that the proposed methodology has relatively simple steps. 1. Select the desired reference variables (RV) 2. If necessary predict the RV that are not available at the target site 3. Calculate the distance between the RV 4. Form the neighborhood based on the previous distance 5. Fit a regional model 6. Predict the target site As mentioned the points 3-6 are common procedure in RFA. The authors agree that the 3 main steps in page 6 could be decomposed in more direct steps as above. In a revised version of the manuscript the results section will provide clearer indication of which step the discussion takes place. The authors did not intend to impose a specific procedure for choosing the RV in its methodology. In the result section, the choice is made to adopt a backward stepwise selection procedure. This procedure is commonly used in regression modeling to select the explanatory variables. The effect of a RV on the final prediction is not straightforward and depends of the "interaction" with the other variables. For instance, two RV can be very well predicted, but if both contained the same information, one will be rejected by the procedure. In the present situation, the

LSK is not well predicted, but it still appears to bring few information that is not contained in the other RV, which improves the final prediction. Therefore, LSK is included. In a revised version of the manuscript the stepwise procedure will be introduced directly in the methodology section. (2.2) Also, the authors did not show some details such as the additional translation which necessary to avoid numerical difficulties of LSK and LKT due to negative values (P10 L19). LSK and LKT are skewed but not positive distribution. In the revised version of the manuscript the following sentences will be modified to provide a mathematical formulation of the actual transformation (P10 L19). "These reference variables are transformed to obtain near symmetric distribution and standardized to obtain zero mean and unit variance. More precisely the transformation for L1 and LCV is the logarithm and for LSK and LKT, the transformation is , where is the minimum of the reference variables. The translation is necessary as LSK and LKT are not positive distribution." (3.1) Although the authors introduced a complicated methodology, they did not make enough efforts to clarify the description of the results; such as confusing explanation of Fig. 4 (P12 L8 – 17) (e.g., why 80 sites? in P12 L12), and unclear Fig. 5 and its explanation (P12 L26 – 33). The Figure 4 presents 4 cross-validation criteria in respect of the different possible neighborhood sizes for calibration. The main point of this approach is to show that selecting a size implies a trade-off between the different criteria. Additionally to the Figures 4 and 6, the author will include in a revised version of the manuscript the following summary table, which may be an easier and more conventional way of presenting the same results. Model RRMSE NHS AHM ARE Index-Flood ROI 46.2 86.5 72.8 57.3 CCA 45.4 86.2 41.7 42.9 RVN-LM 45.0 87.1 14.5 36.9 RVN-HYB 40.1 86.2 16.5 43.1 Regression-based ROI 44.9 86.9 72.8 64.7 CCA 45.2 86.4 40.7 26.7 RVN-LM 42.6 87.3 14.2 34.3 RVN-HYB 42.3 86.2 21.2 51.1 Best criteria in bold The author agree that best NHS for RV-HYB at 80 sites is surprising. However, Figure 4a and 4b mostly shows that regionalization is not very useful in terms of NHS, but it is important in terms of RRMSE, which is the reason why the authors use RRMSE as a calibration criterion. In Figure 5, small changes in average can be difficult to assess by the naked eye. The red lines are smooth curves

Interactive
comment

fitted between the residuals of the two models and are there to indicate which models in average have locally lower residuals. It can be seen that for the residuals above approximately 0.2 ( i.e. 20% of the observed values) the red line is distinctively under the bisector (upper right corner) and this is true for several points. This means the sites that are overestimated are in average less overestimated by the RVN-HYB method. Actually, 8 out of 9 residuals located at the most right are better predicted by the RVN-HYB model. In a revised version of the manuscript the meaning of Figure 5 and the red line will be better explained and the arguments above will be added. I invite to see also the answer to the comments of the another reviewer where some aspects of the figure 5 are discussed in more details. (3.2) Furthermore, the presentation of the results of the regression-based model needs improvements to be clearer (P12 L35 – P13 L10). I recommend using the simple Q-Q plot to assess the compared methods regarding the estimation of regional flood quantile. The authors want to highlight the fact that the description of the steps of the regression-based model in the result section is voluntarily short because they are the same as the index-flood model. Based on the new 6 steps proposed in major comments (2.1), the only modified steps is the steps 5-6, which consist simply to fit a common linear model on the at-site quantile. A sentence will be added in the revised version of the manuscript to mention that more directly. The authors agree with the reviewer and will include a QQ plots in a revised version of the manuscript to improve the analysis of the regression-based model. (3.3) Also, the results should contain numerical tables to quantitatively clarify the differences between the considered methods. The authors can find a close example for the presentation of such results in the reference Gado and Nguyen (2016). Finally, comparing the results of the index flood and the regression methods would be valuable here. The summary table proposed by the reviewer is presented in major comment (3.1). This table will be used in the revised version of the manuscript to briefly compare the index-flood and the regression-based model. However notice that this comparison will be coherent with what is already known in the literature, i.e. that both models performed similarly, except that the index-flood have more coherent quantiles regarding

the different return periods. See for instance: GREHYS, 1996. Presentation and review of some methods for regional flood frequency analysis. Journal of Hydrology 186, 63–84. GREHYS, 1996. Inter-comparison of regional flood frequency procedures for canadian rivers. Journal of hydrology(Amsterdam) 186, 85–103. Haddad, K., Rahman, A., 2012. Regional flood frequency analysis in eastern Australia: Bayesian GLS regression-based methods within fixed region and ROI framework – Quantile Regression vs. Parameter Regression Technique. Journal of Hydrology 430–431, 142 – 161. doi:10.1016/j.jhydrol.2012.02.012

Minor (1) P1 L16 – 17. Which properties does the hydrological information in Regional Frequency Analysis enforce for a group of gauged stations? I suggest to add "desired properties".

The modification will be done in a revised version of the manuscript.

(2) P1 L18. Ungauged sites can be defined by site characteristics in the neighborhood delineation methods (e.g., ROI). Therefore, there is no a challenge for using neighborhoods in RFA regarding the unavailable hydrological information at ungauged sites. The study of Oudin et al. (2010) shows that pooling sites together based on the similarity between the physiographical variables does not necessarily lead to the same group of sites as if hydrological similarity was considered. Their study reports a case of 60% overlapping sites. In other words, it means that if you would like to pool together sites based on hydrological similarity to extrapolate the behavior of another site, but that you substitute it by physiographical information, then 40% the identified sites would not be the ones that will have been chosen if the hydrological information was available. Therefore by "challenge" the authors mean that physiographical cannot replace the missing hydrological information completely. In the revised version of the manuscript, the following sentence will be modified: "A challenge for using neighborhoods in RFFA is that hydrological information is not available at target locations and it cannot be completely replaced by the available physiographical information."

Oudin, L., Kay, A., Andréassian, V., Perrin, C., 2010. Are seemingly physically similar catchments truly hydrologically similar? Water Resources Research 46, n/a–n/a. doi:10.1029/2009WR008887

(3) P1 L23. The regional frequency analysis can be applied for flood or extreme rainfall or any other extreme events. Hence, it should be stated that the case study is for regional flood estimation. The authors agree to specify the scope of the study. In the revised version of the paper, the Acronym RFA for "Regional Frequency analysis" will be changed for RFFA for "Regional Flood Frequency Analysis".

(4) P2 L21. "the distance between hydrological variables". The distance is between locations not variables. The authors disagree with the reviewer. Different notions of distances can be constructed by considering different spaces. The distance between locations is the geographical distance and represent a measure of the separation of two points in the geographical space. Similarly, the hydrological distance is defined as the separation between two sites in the space of the hydrological variables. The sentence below will be modified accordingly in a revised version of the manuscript (P2-L20): "To identify the most similar gauged sites in terms of hydrological properties, a notion of distance is needed to evaluate the proximity, or relevance, of each gauged site to the target location and identify the most hydrologically similar gauged sites."

(5) P3 L4. "as an estimation model" The modification will be done in revised version of the manuscript.

(6) P9 L4. Please, define NH in equation 14. In Eq. (14), correspond to the product of the variable N and H, which are defined. corresponds to the number of gauged sites in the neighborhoods and is the heterogeneity measure in Eq. (13) calculated on all the available gauged sites. In a revised version of the manuscript a product symbol will be added to Eq. (14) to clarify that it is the product of two separate variables.

(7) P9 L14 – 15. How can a regression model fitted on two different neighborhoods, for the same target location, obtain identical values? A simple illustration would be to

consider sites with value: {1,2,3,4,5}. Two delineation methods lead to the group {1,3,5} and {2,3,4}. Here, both groups have predicted value 3 as their mean, but the first group has a variance of 4 and the second has a variance of 1. The following sentence will be modified in the revised version of the manuscript to clarify that "identical values" stands for "similar predicted values". "Notice that a regression model fitted on two different neighborhoods (for the same target location) can lead to very similar predictions, but with different levels of uncertainty."

(8) P10 L2. I don't believe that 15 years of data are enough to get statistically reliable results, why did authors choose 15 years as the minimum time series used in the study. This choice has not been made in the present study. As indicated in P10-L3: "Only a brief description of the data and the at-site frequency analysis is provided since the elements were presented in detail in previous studies (e.g., Chokmani and Ouarda, 2004)." Notice also that 15 years is a minimum. In general the time series are longer and the average is actually 31 years. The average length information will be added to a revised version of the manuscript.

(9) P10 L3. I think you should have at least a map showing the locations of the selected stations in the case study (Quebec). The authors agree with the reviewer and a map will be added to the revised version of the manuscript.

(10) P10 L7. Using the maximum likelihood for parameter estimation with small time series (e.g., 15 years) may cause convergence problems, I would recommend using L-moments instead. As mentioned in the minor comment (8) the at-site frequency analysis was performed and validated in previous studies, where the necessary pre-caution were taken to assure reliable estimates. Moreover, notice the use of the terms "including" and "In general" in the following sentences of the present manuscript to underline that only a brief description of the full methodology is provided: "The at-site distributions are selected among several families including: generalized extreme values (GEV), Pearson type III (P3), generalized logistic (GLO) and log-normal with 3 parameters (LN3). In general, the estimation of the at-site distribution was achieved

by maximum likelihood and the final choices of distributions are based on the Akaike information criterion."

(11) P11 L16. What does HYB denote for in "RVN-HYB"? The acronym "HYB" stand for hybrid, because hydrological and physiographical variables are used as reference variables. The following sentences will be added to the revised version of the manuscript to clarify this point (P10-L15). "The first group is based on L-moments only and the second is based on the combination of L-moments and site-characteristics. The acronym LM for L-moment and HYB for hybrid are used to differentiate them."

(12) P11 L19. "One of the objectives of RFA is to identify a proper family of distributions from regional information" This is not an objective of the RFA. I suggest to write one of the main steps. In a revised version of the manuscript the sentence will be modified as follows: "One of the main steps in RFFA is to identify a proper family of distributions from regional information, which is achieved here by analyzing the distribution of the gauged sites inside a neighborhood."

(13) P11 L23. Please, define here the Q(r) as the regional quantile. In the revised version of the manuscript the sentence will be changed as follows: "In this model, the quantile corresponding to a return period at a target location , where is the index-flood."

Please also note the supplement to this comment:
http://www.hydrol-earth-syst-sci-discuss.net/hess-2016-123/hess-2016-123-AC2-supplement.pdf

---

## Referee Comment (RC3) · T. Gado (Referee) · 7 Jun 2016

General Comments:

The present manuscript investigates the utilization of hydrological information in Regional Frequency Analysis (RFA) in order to improve homogenous properties of neighborhoods and then improve regional flood estimation. I think that the contribution of the study is relevant, and the authors have adequately responded to most of the comments of the reviewers. However, I still have one major comment, two "unfinished" specific comments regarding the first review, and one additional minor comment.

Major comments:

The authors support the idea of using the estimation of hydrological variables, instead of site characteristics, to delineate homogeneous regions. Yet, the estimation of hydrological variables is based on subjective selections of site characteristics and subject to model errors. Moreover, since, homogeneity tests (e.g. Hosking and Wallis 1997) are generally based on hydrological variables (e.g., L1, LCV), these variables should not be used in delineating homogeneous regions. In other words, the same information should not be used for both delineating the homogeneous regions and testing the homogeneity of such regions. And this clarifies the good results regarding the improvements of the homogenous properties (i.e., the results of AHM and ARE) of the resulted neighborhoods by the new method, while the improvements in the results of the regional flood estimations are insignificant (i.e., the results of RMSE and NHS).

Specific comments:

1. P2 L21. "the distance between hydrological variables". The distance is between locations not variables. I guess that the authors misunderstood this comment. I am aware that the distance between two locations can be geographical distance or hydrological distance. However, the distance still should be between locations not variables, otherwise, what is the distance between the two hydrological variables L1 and LCV?

2. P11 L23. Please, define here the $Q(r)$ as the regional quantile. The authors defined the $Q(r)$ on P11 L29 but it still needed to be defined immediately after the equation in P11 L23.

Minor comments:

The second part of the title of figure 3: (b), (c), and (d) Regional L-moments based on the 15 nearest gauged sites for 3 selected target locations.

References

Hosking, J.R.M., Wallis, J.R., 1997. Regional frequency analysis: an approach based on L-moments. Cambridge Univ Pr.

---

## Referee Comment (RC4) · Anonymous Referee #1 · 17 Jun 2016

In the first-round review, my major concerns were related to the "complexity" of the proposed procedure with respect to the problem under analysis and the obtained results. The authors, in their reply, have deeply discussed all the points raised in the review by providing interesting comments and some new numerical results. Although I am not particularly fond of this approach in general, I recognize that these comments can give a considerable improvement to the contents of paper.

A revised manuscript should thus include the comments and develop the points reported in the reply. Moreover, I would like to underline (as already noted in the first comment), that a substantial review of the text organization is still necessary to make the manuscript easier to understand. In particular, the authors should 'help' the reader

to follow all the (many) steps of the procedure.

---

## Author Comment (AC3) · 28 Jun 2016

**1  Major**

**1.1  **The authors support the idea of using the estimation of hydrological variables, instead of site characteristics, to delineate homogeneous regions. Yet, the estimation of hydrological variables is based on subjective selections of site characteristics and subject to model errors.**

The authors want to clarify that the idea itself of using hydrological variables, called here reference variables (RV), was not proposed in the present study. The traditional CCA method has suggested already to delineate homogenous regions using flood quantiles as RV. More precisely, the idea supported in the present paper is that a larger class of RV could be considered as well as different estimation methods.

The authors agree that there is uncertainty in the RV, but it does not represent an additional uncertainty. The ROI method has implicitly these uncertainties too. If the predicted RV can be taken as the average of the ROI neighborhoods, as it is done with the index-flood model, then these predicted RVs are not predicted without error. This fact is illustrated in Figures 3b,c,d where the average LSK and LCV are not centered at the target location. It is actually an advantage of the proposed method to explicitly model that uncertainty. A mention of this advantage will be added in section 3.1 of the revised version of the manuscript, where the concept of RV is introduced (P7 L5).

[Figure]

**1.2** **Moreover, since, homogeneity tests (e.g. Hosking and Wallis 1997) are generally based on hydrological variables (e.g., L1, LCV), these variables should not be used in delineating homogeneous regions. In other words, the same information should not be used for both delineating the homogeneous regions and testing the homogeneity of such regions.**

The authors understand the concern of the reviewer and agree that in the framework proposed by Hosking and Wallis (1997) the same hydrological variables could not have been used for delineating and testing the homogenous regions. However, the present methodology does not perform any homogeneity tests. The criteria used for selecting the size of the neighborhood is the RRMSE and is based on cross-validation, which tends to optimize the prediction corresponding to a specific return period. Consequently, the L-moments are not the variables used in the calibration of the neighborhood.

**1.3** **And this clarifies the good results regarding the improvements of the homogenous properties (i.e., the results of AHM and ARE) of the resulted neighborhoods by the new method, while the improvements in the results of the regional flood estimations are insignificant (i.e., the results of RMSE and NHS)**

The authors thank the reviewer for raising this question, which gives them the opportunity to clarify this point. However, the authors do not agree that the terminology "insignificant improvements" properly describes the results of the present study. The authors have argued inside the answer AC1 of the online discussion that the results in terms of RRMSE are in fact substantial. Briefly, the improvement for the RVN-HYB method in comparison of the ROI method is of 6.1% for the RRMSE criteria and it is shown that the improvements are more important for sites that have larger discrepancies.

Indeed, better performances in terms of AHM and ARE are a direct consequence of using RV, which is the main purpose of the proposed methodology. The authors believe that the important point is not if these criteria are better, but how much better they are. In the revised version of the manuscript the author will discuss in more details the magnitude of this difference:

(P12 L20) "Figures 4c,d present respectively the AHM and the ARE criteria. The AHM criterion indicates that the ROI and the CCA methods have in general lower heterogeneity than the whole dataset, but are outperformed by the RVN-LM and RVN-HYB methods especially for smaller neighborhoods. This quantifies the intuitive assumption that the regional LCV is calculated with less uncertainty when the L-moments are directly considered instead of other reference variables. In particular, the AHM of the ROI method is 72.8% with the optimal neighborhood size of 30. In comparison, the AHM of the RVN-LM method is 14.5% with the optimal neighborhood size of 28 sites, which is considerably lower. Figure 4c shows that the AHM criterion of the RVM-LM method does not reach a similar level to the ROI method until using as much as 120 sites. These results indicate that even for relatively small neighborhoods, the ROI method identifies regions that are only slightly less hydrologically heterogeneous than all sites pooled together. This suggests that, in the present case study, the ROI method has difficulties identifying sites that are similar to the target site in terms of LCV."

**2 Specific**

**2.1 P2 L21. "the distance between hydrological variables". The distance is between locations not variables. I guess that the authors misunderstood this comment. I am aware that the distance between two locations can be geographical distance or hydrological distance. However, the distance still should be between locations not variables, otherwise, what is the distance between the two hydrological variables L1 and LCV?**

The authors would like to thank the reviewer for pointing out this blunder. The authors agree that the formulation needs to be changed and distances remain between locations not between variables. The sentence below will be modified accordingly in the revised version of the manuscript (P2-L20):

"To identify the most similar gauged sites in terms of hydrological properties, a notion of distance is needed to evaluate the proximity, or relevance, of each gauged site to the target location and identify the most hydrologically similar gauged sites."

**2.2 P11 L23. Please, define here the Q(r) as the regional quantile. The authors defined the Q(r) on P11 L29 but it still needed to be defined immediately after the equation in P11 L23.**

The author agrees with the reviewer. The change indicated by the reviewer will be done in the revised version of the manuscript.

**3  Minor**

**3.1  **The second part of the title of figure 3: (b), (c), and (d) Regional L-moments based on the 15 nearest gauged sites for 3 selected target locations.**

The correction will be made in the revised version of the manuscript. The authors reiterate their thanks to the reviewer.

---

## Author Response (AR1)

**Delineation of homogenous regions using hydrological variables predicted by projection pursuit regression**

Martin Durocher, Fateh Chebana, and Taha B. M. J. Ouarda

Manuscript: hess-2016-123

Dear Editor,

Please find herewith the response to all comments made by the two reviewers concerning the manuscript "Delineation of homogenous regions using hydrological variables predicted by projection pursuit regression" in the interactive discussion on the website of Hydrology and Earth System Science (HESS).

The authors gratefully acknowledge the helpful comments that have contributed to the improvement of the paper. Detailed replies to these comments are provided below.

Should you have any questions or require further information concerning this revision, please contact me.

Sincerely,
Martin Durocher

**Reply to Reviewer 1 comments**

The authors are grateful to the reviewer for his comments which contributed to improving the quality of the paper. The authors provide hereafter the answers to the reviewer's comments. Please note that the numbering of the figures has been changed, which explains the difference between the reviewer's comments and the provided answers.

**General comments**

**Reviewer: The article "Delineation of homogeneous regions using hydrological variables predicted by projection pursuit regression" by Durocher et al. describes an improvement of existing techniques for the regional estimation of flood quantiles. The topic is very relevant, but I found the manuscript not completely clear in some parts and with some methodological flaws. While readability (see comment n. 1) can be improved with a revision of the text, some methodological issues would require a complete reanalysis of the work. My main concerns (see comments n.2 and 3) are basically related to the use of a very complex procedure that is not justified by the results. This finding (i.e., that the procedure does not really produce improvements) would be a result itself, but the authors seem to overlook it to support their initial hypotheses. For these reasons, I suggest a rejection of manuscript.**

**Answer:** The authors would like to thank the Reviewer for the thorough review and the constructive comments. In the following we will provide a response to the comments formulated by the reviewer. Please note that the response to the general comments (mainly the justification for a complex model) is in fact available in the response to the major comments below.

**Major comments**

**(1.1)  Reviewer: "The point list on page 6, and in particular the step i) (which is the main focus of the paper) should be supported by a quantitative example to make the procedure easier to understand. For instance, the plots in figure 3 could be used in this part of the manuscript to better describe how the procedure works (and not only from page 11 to comment results)."**

**Answer:** The authors agree that the illustration provided by Figure 4 (initially Figure 3) is useful to understand this step of the methodology. Figure 1 was initially intended to provide a schematic illustration of this step. Figure 1 was improved in the revised manuscript in order to be more similar in its interpretation. The following sentences were modified in agreement with the new figure:

> P7-L19, "Figure 1 illustrates a region with several sites where two neighborhoods are resulting from the RVN method with different predicted centers. The target site is illustrated as a green filled circle and neighborhood is formed of the 10 nearest sites indicated by small empty circles. The other sites are designated by crosses. The red and blue neighborhoods are delineated by circles where the radius is selected to include the 10 nearest sites. The predicted center of the red neighborhood is closer to the target site. Consequently, it can be seen that except from one site, the same sites as the target neighborhood are included (empty circles). On the other hand, the blue neighborhood has a predicted center further to the target site and hence a lower proportion of the sites truly

closer to the target are found. It shows the importance of correctly predicting neighborhood centers in order to identify sites that are truly similar to the target site."

**(1.2) Reviewer: "Moreover, step i) seems a kind of "preliminary" regionalization of the L-moments of the target site. Such L-moments are then used to support the delineation of the region. Why such preliminary estimates cannot be directly used in the prediction of flood quantiles? This point should be discussed by the authors, highlighting the possible differences with the direct estimation of the quantiles based on preliminary L-moments."**

**Answer:** In a situation where the target distribution is assumed to be known (i.e. the same family of distribution is assumed everywhere) the preliminary prediction could be used indeed to deduce the parameter of the target distribution from the L-moments. However, when the distribution is unknown, the L-moments cannot be used directly to estimate the flood quantile without specifying a family of distributions. This is achieved here by using the information of the relevant neighborhoods.

Moreover, the approach of using the preliminary L-moments is not generally applicable to the present RVN methodology as it fixed the reference variables, while the present methodology includes a stepwise procedure to perform this selection. The following sentences are added to the revised manuscript:

> P12-L12, "At this point, the steps 1-4 of the RVN methodology are performed and the neighborhoods are identified. Notice that for the RVN-LM method, the reference variables include the first three L-moments, which could be used as a moment estimator to deduce the target distribution. This approach is, however, not generally applicable to the present methodology as the reference variables are selected by a stepwise procedure. Moreover, it is necessary to identify a proper family of distributions from regional information, which is achieved here by analyzing the distribution of the gauged sites inside the neighborhoods."

**(2.1) Reviewer: "Figure 5 tells me that there is no significant difference between the methods, so I would use the simpler one. Of course, by computing an error metric over the whole set of residuals one may obtain slightly better performances of the RVN models (but this is not reported; a summary table would be appreciated). "**

**Answer:** Notice that two error metrics are actually used in the present study: RRMSE and NHS. The authors agree that an additional table would help summarize more directly the information provided in the text and in Figures 5 and 8 (initially 4 and 6) for the calibrated model. In this sense, Table 1 below is added to revised manuscript:

**Table 1: Evaluation criteria for the RVN method for optimal neighborhood sizes.**

| | Model | Size | RRMSE | NHS | AHM | ARE |
|---|---|---|---|---|---|---|
| **Index-Flood** | | | | | | |
| | ROI | 30 | 46.2 | 86.5 | 72.8 | 57.3 |
| | CCA | 28 | 45.4 | 86.2 | 41.7 | 42.9 |
| | RVN-LM | 29 | 45.0 | **87.1** | **14.5** | **36.9** |
| | RVN-HYB | 24 | **40.1** | 86.2 | 16.5 | 43.1 |
| **Regression-based** | | | | | | |
| | ROI | 30 | 44.9 | 86.9 | 72.8 | 64.7 |
| | CCA | 28 | 43.5 | 86.1 | 41.7 | **30.6** |
| | RVN-LM | 39 | 41.7 | **87.6** | 17.9 | 39.8 |
| | RVN-HYB | 24 | **39.5** | 86.2 | **16.5** | 42.5 |

Best criteria in bold

**(2.2) Reviewer: "The authors state on page 12, line 25 onwards, that improvement is effective for sites with largest discrepancies. This seems not true except for two point in figure 5a and one point in figure 5b (all the points in the bottom-left corner of each panel). Figures 5c and 5d show points equally distributed around the bisector also in the bottom-left corner. Hence, is the complexity of the RVN model justified by a so small performance improvement?"**

**Answer:** The authors want to make the precision that the comment "improvement is effective for sites with largest discrepancies" applied only to the relative residuals as this comment is made while describing Figures 6a and 6b (initially 5a,b). To improve clarity, the term "largest relative discrepancies" is used in the revised version of the manuscript:

> P13-L31, "However, Figures 6a,b show that the RVN-HYB specifically improves the prediction of the sites with the lowest and largest relative discrepancies as the red line is clearly located under the $y = x$ lines, which explains the improved RRMSE in Table 1."

The additional complexity of the RVN method in comparison to the traditional ROI method concerns only the preliminary step, where the missing hydrological information is substituted by predicted values. The question is thus if the addition of the preliminary step is justified. Notice that RRMSE and NHS are cross-validated criteria and hence they are criteria that "penalize" excessive complexity in the prediction. Consequently, a better RRMSE implies that a model with a higher criterion truly brings additional information on the quantiles. The overall improvement in terms of RRMSE for the RVN-HYB method with respect of the ROI and CCA methods is respectively 6.1% and 5.3% (see Table 1). Although of moderate amplitude, the authors believe that these improvements are nonetheless important as several references publications in the field had been published with improvements in the range of 5%. The following sentences are modified in the revised manuscript:

> P13-L11, "Hence, the calibrated models are set according to the RRMSE criterion and are represented by circles in Figure 5 and are summarized in Table 1. RVN-HYB, with a RRMSE of 40.1% outperforms the other methods. In particular, a difference of 6.1% and 5.3% is observed respectively with the traditional ROI and CCA methods."

The authors agree with the reviewer that the red lines in the bottom-left corner of Figures 6a,b are mostly influenced by few points. Nevertheless, in panel a, the 4 largest relative discrepancies are better predicted by RVN-HYB and the two best relative improvements are of 77.2% and 68.5%. In these figures, small changes in average can be difficult to assess by the naked eye. The red lines are smooth curves fitted

between the residuals of the two models, and are there to indicate which models in average have locally lower residuals. It can be seen that for the residuals approximately above 0.2 ( i.e. 20% of the observed values) the red line is distinctively under the bisector (upper right corner) and this is true for several points. This means that the sites that are overestimated are on average less overestimated by the RVN-HYB method. Actually, 8 out of the 9 residuals located at the most right are better predicted by the RVN-HYB model. Similar behavior is also noticeable in upper-left corner of Figure 5b.

In the revised version of the manuscript the following lines were modified to better explain the implication of the result shown in Figure 6:

P13-L26, "As mentioned in section 4.2, previous studies have identified few problematic stations in the considered dataset. Figure 6 presents the residuals between different methods. As it may be difficult to see small improvements by uniquely observing points around the $y = x$ lines, the visualization of Figure 6 is helped by adding a flexible fit of the point cloud, using a standard smoothing spline approach. The resulting red lines indicate, in average, if close to $x$ the residuals are lower for one of the two methods. In general, the points associated to the largest relative discrepancies are close to the $y = x$ line, which indicates that the sites that are difficult to predict are essentially the same for all methods. However, Figures 6a,b show that the RVN-HYB specifically improves the prediction of the sites with the largest relative discrepancies as the red line is clearly located under the $y = x$ lines (left and right), which explains that this method leads to the best RRMSE. On the other hand, Figures 6c,d demonstrate that at the logarithmic scale, the RVN-LM method achieved predicted values that are mostly similar to the ROI and CCA methods, which explains the similarity of the NHS criteria for all the compared methods.

The present case study is an example of a region where some sites are problematic for likely any method. In practice, the residuals are not known, consequently we do not know if the target sites of interest will be "problematic" or not. Globally, what Figure 6a indicates is that the RVN-HYB model is more robust (in a certain way), because for the sites that are well predicted by simpler models, such ROI, RVN-HYB will perform in average similarly. However, if the target site is predicted less accurately, the RVN-HYB model will (in average) be better in terms of RRMSE. Consequently, the overall gain may seem of moderate magnitude, but for some problematic stations the gain could be more substantial. In particular, the red lines in the left part of Figure 6a appears mostly influenced by two points, but the two improvements are of 77.2% and 68.5%, which is considerable."

**(3) Reviewer: "On page 10, line 29, the nonlinear relationship between the (transformed) predictors and the (transformed) L-moment is mentioned and the authors say that it is shown in figure 2. This non-linear relationship would justify the use of a spline interpolator, but actually this is a questionable point. In fact, figure 2 tells a different story. Panel a clearly show a linear relationship (in the transformed variables; this is expected as often the mean value can be linearized with log transformations). In the b, c and d panels there is a much larger scattering, which does not allow to identify a clear complex pattern, even if all the plots show an increasing trend. Looking at the scatter plots I believe that most of the people would adopt a simple linear regression (said with 2 parameters) which is much more stable and robust. My personally idea is that the choice of the authors is not justified and that a linear model should be at least compared to the spline interpolator."**

**Answer:** The authors understand the concern of the reviewer in the use of the PPR in Figure 3 (initially Figure 2). Comparison with linear model are made. For L1 and LSK the difference between the NHS criteria is about 1 %, which the authors agree is very small. However, the gains for the LCV and LKT are respectively of 5.1% and of 7.6%, which is substantial. Notice that the NHS is a cross-validation criterion and hence this result does not represent a form of overfitting in favor of the PPR as the predictions are obtained without the use of the predicted sites. Unnecessary complexity is then «penalized» by such criteria. The fitting of LCV and LKT are cases of mild nonlinearity, but such mild nonlinearity can be adjusted here because 151 is a reasonable number of sites. The authors agree that the situation would be different if for instance only 20 sites where available. Therefore, in the present case study, it is not true that linear models are more stable and robust than PPR as they are assessed by cross-validation. The following changes are made to the revised manuscript:

> P11-L26, "Figure 3a shows a strong linear relationship between L1 and the predictor $\alpha'\mathbf{X}$. Conversely, Figures 3b,c,d show mild nonlinearity and hence indicate the need for more flexible models, such as PPR. The predictive performances of the reference variables are evaluated by the NHS criterion with values 91.5%, 33.3%, 6.7% and 55.7% respectively for L1, LCV, LSK and LKT. These results show that L1 is accurately predicted by the site characteristics, while a poor fit is associated to LSK. Indeed, Figure 3c suggests that apart from a few sites on the right of the curve, LSK appears not highly related to the predictor $\alpha'\mathbf{X}$. In comparison, linear models applied on the same reference variables lead to NHS criterion: 90.9%, 28.2%, 7.8% and 48.1% respectively. Remark that NHS criterion is calculated by cross-validation, consequently even though the improved performances by the PPR method appear moderate this represent true fitting improvements."

**Minor comments**

**(1) Reviewer: "P11 L5-7 I found quite strange that the L-kurtosis performs much better that the L-skewness as in general the prediction ability deteriorates with increasing order of L-moments. The authors should investigate in more detail this issue."**

**Answer:** The authors agree with the reviewer that L-skewness is expected to be, in general, better predicted than the L-kurtosis. The data have been investigated for data manipulation errors and nothing have been found. It appears to be a legitimate exception to the rule.

**(2) Reviewer: "P7 L6 Please, give a more detailed description of "true neighborhood" meaning."**

**Answer:** To clarify the term "true neighborhood" the following lines are modified in the revised manuscript:

> P7 L16, "If the hydrological variables $\mathbf{t}_0$ were known at the target location, the distance $h_i$ would be available and the neighborhood that truly regroups the most hydrologically similar sites to the target location can be identified. However, in practice this true neighborhood is unknown. Using instead the estimate $f(\mathbf{x}_0)$ has the effect that some sites are falsely suggested as more hydrologically similar than other sites."

**(3) Reviewer: Figure 5. Not clear which kind of information is provided by the "smooth fitting of the residuals". Also in this case, the smooth fitting seems too complex tool which does not add any further information.**

**Answer:** As briefly discussed in the major comment (2.2), at the proximity of a point say (X,X) in Figure 6 (initially Figure 5), several points maybe under or above the bisector, which makes it difficult to see small advantages for a given method. The smooth fitting, provided by the red lines, indicates specifically the average between the residuals of the two models in the proximity of a point X. The authors agree with the reviewer that parsimony is important. The smoothing splines procedure used here as a visual guide is widely available in most numerical software's and the "degree of complexity" is controlled by the generalized cross-validation criteria (GCV), which is a widely used criterion to avoid overfitting. Hence, with the information from 151 sites, the authors believe that the smooth fitting is not "too complex", but it is simply "as complex" as the residuals suggest it. The following precisions are added to the revised manuscript:

> P13 L27, "As it may be difficult to see small improvements by uniquely observing points around the $y = x$ lines, the visualization of Figure 6 is helped by adding a flexible fit of the point cloud, using a standard smoothing spline approach. The resulting red lines indicate if close to $x$ the residuals are in average lower for one of the two methods."

**Reply to Reviewer 2 comments**

The authors are grateful to the reviewer for his comments which contributed to improving the quality of the paper. The authors provide hereafter the answers to the reviewer's comments. Please be aware that the numbering of the figures has change, which explained difference between the reviewer's comments and the answers provided by the authors.

**Major comments**

**(1) Reviewer: The literature is not complete and does not state what other researchers have done in order to improve the flood estimation at ungauged sites. So, the authors should improve the description of the existing literature on the topic investigated. In particular, the manuscript should elaborate a little bit better on the evolution of the ROI method as the study focuses on the neighborhood approach for homogenous region delineation.**

**Answer:** The authors would like to thank the Reviewer for the thorough review and the constructive comments. In the following we will provide a response to the comments formulated by the reviewer.

The authors agree with the reviewer and the revised version of the manuscript includes an improved description of the existing literature on ROI. Notice, however, that much effort in the recent literature on the ROI method deals with the problem of estimating the model by generalized least squares to account for different aspects, which is not the problem addressed in the present study. The main focus is more about the improvement of the ideas behind the CCA method, for which the recent developments are included in the introduction. The sentences below are added to the revised manuscript:

> P2-L24, "The traditional approach, based on the distance between site characteristics, is commonly referred to as the Region of Influence (ROI) model (Burn, 1990), which received a particular attention in the hydrological literature. The focus was mainly on the estimation of the model parameters, where for instance the generalized least-squares were used to account for unequal variability in the at-site estimations (e.g. Griffis and Stedinger, 2007; Stedinger and Tasker, 1985) and to deal with the presence of spatial correlation (e.g. Kjeldsen and Jones, 2009). "
>
> Kjeldsen, T.R., Jones, D.A., 2009. An exploratory analysis of error components in hydrological regression modeling. Water Resour. Res. 45, n/a–n/a. doi:10.1029/2007WR006283
>
> Griffis, V., Stedinger, J., 2007. The use of GLS regression in regional hydrologic analyses. J. Hydrol. 344, 82–95.
>
> Stedinger, J., Tasker, G., 1985. Regional hydrologic analysis: 1. Ordinary, weighted, and generalized least squares compared. Water Resour. Res. 21, 1421–1432. doi:10.1029/WR021i009p01421

**(2.1) Reviewer: The methodology is blurred, difficult to follow, and contains some odd judgements. For instance, LSK was maintained because it is associated to better predictive performance, however, it is poorly predicted by the site characteristics (P11 L5 – 12).**

**Answer:** The authors believe that the proposed methodology has relatively simple steps. The authors agree that the 3 initial main steps could be decomposed in more direct steps as above. In the revised version of the manuscript, the results section provides clearer indications of which step the discussion deals with.

The authors did not intend to impose a specific procedure for choosing the reference variables in the methodology. In the result section, the choice is made to adopt a backward stepwise selection procedure. This procedure is commonly used in regression modeling to select the explanatory variables. The effect of a reference variable on the final prediction is not straightforward and depends of the "interaction" with the other variables. For instance, two reference variables can be very well predicted, but if both contain the same information, one will be rejected by the procedure. In the present situation, the LSK is not well predicted, but it still appears to bring few information that is not contained in the other RV, which improves the final prediction. Therefore, LSK is included.

The part below is changed in the revised manuscript:

P6-L21, "The general procedure can be described by the steps below:

1. Select the reference variables

2. If necessary, predict the reference variables that are not available at the target site

3. Calculate the distance between the reference variables

4. Form the neighborhood based on the previous distance

5. Fit a regional model on the neighborhood

6. Predict the target site

In step 1, the selection of a set of the reference variables can be subjective and depend on the problem at hand. In the present study, backward stepwise selection procedure is considered to remove from an initial set of references variables those that are not contributing to the prediction power of the model. This selection procedure is more objective and depends on performance criteria that will be described in section 3.2.

Step 2 is required only if some reference variables are unknown at the target sites, otherwise, if a target location designated by $i = 0$, the radius of the neighborhood used in step 3 can be computed as $h_i = d(\mathbf{t}_i, \mathbf{t}_0)$ where $d$ is a metric and $\mathbf{t}_i' = (t_{i,1}, \ldots, t_{i,q})$ are the reference variables of the $i$th site. For simplicity, the Euclidian metric $d$ is considered throughout the present study, but other metrics or dissimilarity measures could be employed as well. In particular, the Mahalanobis distance, the weighted distance and the depth function could be considered (Chebana and Ouarda, 2008; Cunderlik and Burn, 2006; Ouarda et al., 2000).

If some hydrological information is unavailable at the target location, the estimation of the hydrological reference variables is necessary to produce an estimate $\mathbf{t}_0 = f(\mathbf{x}_0)$ in step 2 from site characteristics $\mathbf{x}_0$ at the target location. This substitution leads in step 3 to the distance $h_{(i)} = d[\mathbf{t}_i, f(\mathbf{x}_0)]$, which may be seen as an approximation of the true distance $h_i$. This study considers PPR models in order to fit every hydrological reference variable as described in section 2.3. The motivations for adopting PPR are that it does not require a prior delineation of regions, it accounts for nonlinear relationships, it has good predictive performances and it leads to a straightforward interpretation of the reference variables when a few directions $\alpha_k$ are necessary (Durocher et al., 2015)."

**(2.2) Reviewer: Also, the authors did not show some details such as the additional translation which necessary to avoid numerical difficulties of LSK and LKT due to negative values (P10 L19).**

**Answer:** The distributions of LSK and LKT are skewed but not positive. Hence the logarithm is used on a translated variable instead. In the revised version of the manuscript, the following sentences are modified to provide a mathematical formulation of the actual transformation:

P11-L7, "These reference variables are transformed and standardized to obtain zero mean and unit variance. More precisely, the transformation for L1 and LCV is the logarithm and for LSK and LKT, the transformation is $\log(x - m_x + 1)$, where $m_x$ is the minimum of the reference variables."

**(3.1) Reviewer: Although the authors introduced a complicated methodology, they did not make enough efforts to clarify the description of the results; such as confusing explanation of Fig. 4 (P12 L8 – 17) (e.g., why 80 sites? in P12 L12), and unclear Fig. 5 and its explanation (P12 L26 – 33).**

**Answer:** Figure 5 (initially Figure 4) presents the 4 cross-validation criteria with respect to the different possible neighborhood sizes. The main point of this approach is to show that selecting a calibrated size implies a trade-off between the different criteria. Additionally, to the Figures 5 and 8, the authors include Table 1 in the revised manuscript that summarized these criteria.

The authors agree that best NHS for RV-HYB at 80 sites is surprising. However, Figures 5a and 5b mostly show that regionalization is not very useful in terms of NHS, but it is important in terms of RRMSE. The authors use RRMSE as a calibration criterion.

In Figure 5, small changes in average can be difficult to assess by the naked eye. The red lines are smooth curves fitted between the residuals of the two models and are there to indicate which models in average have locally lower residuals. The explanation below is added to the revised manuscript:

P13 L26, "As mentioned in section 4.2, previous studies have identified few problematic stations in the considered dataset. Figure 6 presents the residuals between different methods. As it may be difficult to see small improvements by uniquely observing points around the $y = x$ lines, the visualization of Figure 6 is helped by adding a flexible fit of the point cloud, using a standard smoothing spline approach. The resulting red lines indicate if close to $x$ the residuals are lower in average for one of the two methods. In general, the points associated to the largest relative discrepancies are close to the $y = x$ line, which indicates that the sites that are difficult to predict are essentially the same for all methods. However, Figures 6a,b show that the RVN-HYB specifically

improves the prediction of the sites with the largest relative discrepancies as the red line is clearly located under the $y = x$ lines (left and right), which explains that this method leads to the best RRMSE. On the other hand, Figures 6c,d demonstrate that at the logarithmic scale, the RVN-LM method achieved predicted values that are mostly similar to the ROI and CCA methods, which explains the similarity of the NHS criteria for all the compared methods.

The present case study is an example of a region where some sites are problematic for likely any methods. In practice, the residuals are not known, consequently we do not know if the target sites of interest will be "problematic" or not. Globally, what Figure 6a indicates is that the RVN-HYB model is more robust (in a certain way), because for the sites that are well predicted by simpler models, such ROI, RVN-HYB will perform in average similarly. However, if the target site is predicted less accurately, the RVN-HYB model will (in average) be better in terms of RRMSE. Consequently, the overall gain may seem of moderate magnitude, but for some problematic stations the gain could be more substantial. In particular, the red lines in the left part of Figure 6a appears mostly influenced by two points, but the two improvements are of 77.2% and 68.5%, which is considerable."

**(3.2) Reviewer: Furthermore, the presentation of the results of the regression-based model needs improvements to be clearer (P12 L35 – P13 L10). I recommend using the simple Q-Q plot to assess the compared methods regarding the estimation of regional flood quantile.**

**Answer:** The authors want to highlight that the description of the steps of the regression-based model in the result section is voluntarily short because they are the same as the index-flood model. Based on the new 6 steps of the methodology (see comments 2.1), only the steps 5-6 change, which consists to fit a common linear model on the at-site quantile. The sentences below are added to the revised manuscript:

P14-L8, "Prediction of Q100 at the target location is also performed by the regression-based model using the same delineation methods as with the index-flood model, but with potentially different calibration values for the neighborhood sizes. Consequently, the description of steps 1-4 (in section 3.1) are identical to those of the index-flood approach and are not repeated here."

The authors agree with the reviewer and QQ plots (Figure 7) are included in the revised version of the manuscript to improve the analysis of the regression-based model:

P14-L10, "The fit of the regression-based model is graphically assessed in Figure 7 by Quantile-Quantile plots. It is showed that for all delineation approach the regression-based models correctly predict the flood quantile Q100 at target."

**(3.3) Reviewer: Also, the results should contain numerical tables to quantitatively clarify the differences between the considered methods. The authors can find a close example for the presentation of such results in the reference Gado and Nguyen (2016). Finally, comparing the results of the index flood and the regression methods would be valuable here.**

**Answer:** The added Table 1 now compared the index-flood and the regression-based model and the following sentences are added to the revised manuscript

P14 L24: "Table 1 provides also a comparison between the performance of the index-flood and the regression-based model. In terms of RRMSE and NHS criterion, the two approaches lead to very

similar results, which is coherent with what it is reported in other studies (GREHYS, 1996a, 1996b; Haddad and Rahman, 2012). Therefore, similar conclusions can be draw from the two approaches. For instance, in both cases, the RVN-HYB leads to the best results in terms of RRMSE."

GREHYS, 1996. Presentation and review of some methods for regional flood frequency analysis. Journal of Hydrology 186, 63–84.

GREHYS, 1996. Inter-comparison of regional flood frequency procedures for canadian rivers. Journal of hydrology(Amsterdam) 186, 85–103.

Haddad, K., Rahman, A., 2012. Regional flood frequency analysis in eastern Australia: Bayesian GLS regression-based methods within fixed region and ROI framework – Quantile Regression vs. Parameter Regression Technique. Journal of Hydrology 430–431, 142 – 161. doi:10.1016/j.jhydrol.2012.02.012

**(4.1) Reviewer: The authors support the idea of using the estimation of hydrological variables, instead of site characteristics, to delineate homogeneous regions. Yet, the estimation of hydrological variables is based on subjective selections of site characteristics and subject to model errors.**

The authors want to clarify that the idea itself of using hydrological variables, called here reference variables (RV), was not proposed in the present study. Implicitly, the traditional CCA method has suggested already to delineate homogenous regions using flood quantiles as reference variables. More precisely, the idea supported in the present paper is that a larger class of reference variables could be considered as well as different estimation methods.

The authors agree that there is uncertainty in the reference variables, but it does not represent an additional uncertainty for the proposed method. The ROI method has implicitly these uncertainties. If the predicted reference variables can be taken as the average of the ROI neighborhoods, as it is done with the index-flood model, then these predicted reference variables are not predicted without error. This fact is illustrated in Figures 4b,c,d of the revised manuscript where the average LSK and LCV are not centered at the target location. It is actually an advantage of the proposed method to explicitly model that uncertainty.

> P7-L28, "The errors related to prediction of the hydrological reference variables suggest that the RVN method may include an additional source of uncertainty, which is not accurate. Indeed, the same source of uncertainty is present among the sites of a neighborhood delineated on the basis of known site characteristics (i.e that the average of the hydrological variables in the neighborhood is not a perfect predictor). This can be seen as an advantage of the RVN method since it directly assesses this source of uncertainty and tries to reduce it."

**(4.2) Reviewer: Moreover, since, homogeneity tests (e.g. Hosking and Wallis 1997) are generally based on hydrological variables (e.g., L1, LCV), these variables should not be used in delineating homogeneous regions. In other words, the same information should not be used for both delineating the homogeneous regions and testing the homogeneity of such regions.**

The authors understand the concern of the reviewer and agree that in the framework proposed by Hosking and Wallis (1997) the same hydrological variables could not have been used for delineating and testing the homogenous regions. However, the present methodology does not perform any homogeneity test. The criteria used for selecting the size of the neighborhood is the RRMSE and is based on crossvalidation, which tends to optimize the prediction corresponding to a specific return period. Consequently, the L-moments are not the variables used in the calibration of the neighborhood.

**(4.3) Reviewer: And this clarifies the good results regarding the improvements of the homogenous properties (i.e., the results of AHM and ARE) of the resulted neighborhoods by the new method, while the improvements in the results of the regional flood estimations are insignificant (i.e., the results of RMSE and NHS)**

The authors thank the reviewer for raising this question, which gives them the opportunity to clarify this point. However, the authors do not agree that the terminology "insignificant improvements" properly describes the results of the present study. The authors have already argued that the results in terms of RRMSE are in fact substantial. Briefly, the improvement for the RVN-HYB method in comparison to the ROI method is of 6.1% in terms of the RRMSE and it is shown that the improvements are more important for sites that have larger discrepancies (See comment 2.2 of the first reviewer).

Indeed, better performances in terms of AHM and ARE are a direct consequence of using reference variables, which is the main purpose of the proposed methodology. The authors believe that the important point is not if these criteria are better, but how much better they are. In the revised version of the manuscript the author added the explanations below:

> P13-L15, "Figures 5c,d present respectively the AHM and the ARE criteria obtained from the considered methods. The AHM criterion indicates that the ROI and the CCA methods have in general lower heterogeneity than the whole dataset, but are outperformed by the RVN-LM and RVN-HYB methods especially for smaller neighborhoods. This quantifies the intuitive assumption that the regional LCV is calculated with less uncertainty when the L-moments are directly considered instead of other reference variables. In particular, the AHM of the ROI method is 72.8% with the optimal neighborhood size of 30. In comparison, the AHM of the RVN-LM method is 14.5% with the optimal neighborhood size of 29 sites, which is considerably lower. Figure 5c shows that the AHM criterion of the RVM-LM method does not reach a similar level to the ROI method until using as much as 120 sites. These results indicate that even for relatively small neighborhoods, the ROI method identifies regions that are only slightly less hydrologically heterogeneous than all sites pooled together. This suggests that, in the present case study, the ROI method has difficulties identifying sites that are similar to the target site in terms of LCV."

**Minor comments**

**(1) Reviewer: P1 L16 – 17. Which properties does the hydrological information in Regional Frequency Analysis enforce for a group of gauged stations? I suggest to add "desired properties".**

**Answer:** The modification is done in the revised version of the manuscript:

> P1-16, "This study investigates the utilization of hydrological information in Regional Flood Frequency Analysis (RFFA) to enforce desired properties for a group of gauged stations."

**(2) Reviewer: P1 L18. Ungauged sites can be defined by site characteristics in the neighborhood delineation methods (e.g., ROI). Therefore, there is no a challenge for using neighborhoods in RFA regarding the unavailable hydrological information at ungauged sites.**

**Answer:** The study of Oudin et al. (2010) shows that pooling sites together based on the similarity between the physiographical variables (i.e. site characteristics) does not necessarily lead to the same group of sites as if hydrological similarity was considered. Their study reports a case of 60% overlapping sites. In other words, it means that if one would like to pool together sites based on hydrological similarity to extrapolate the behavior of another site, but that one substitutes it by physiographical information, then 40% the identified sites would not be the ones that will have been chosen if the hydrological information was available.

Therefore, by "challenge" the authors mean that physiographical cannot replace the missing hydrological information completely. In the revised version of the manuscript, the following sentence :

> P1-L18, "A challenge for using neighborhoods in RFFA is that hydrological information is not available at target locations and it cannot be completely replaced by the available physiographical information."

Oudin, L., Kay, A., Andréassian, V., Perrin, C., 2010. Are seemingly physically similar catchments truly hydrologically similar? Water Resources Research 46, doi:10.1029/2009WR008887

**(3) Reviewer: P1 L23. The regional frequency analysis can be applied for flood or extreme rainfall or any other extreme events. Hence, it should be stated that the case study is for regional flood estimation.**

**Answer:** The authors agree to specify the scope of the study. In the revised version of the paper, the Acronym RFA for "Regional Frequency analysis" is replaced by RFFA for "Regional Flood Frequency Analysis".

**(4) Reviewer: P2 L21. "the distance between hydrological variables". The distance is between locations not variables.**

**Answer:** The authors would like to thank the reviewer for pointing out this blunder. The authors agree that the formulation needs to be changed and distances remain between locations not between variables. The sentence below is modified accordingly in the revised version of the manuscript:

> P2-L19, "To identify the most similar gauged sites in terms of hydrological properties, a notion of distance is needed to evaluate the proximity, or relevance, of each gauged site to the target location and identify the most hydrologically similar gauged sites. However, when the target location is ungauged, this distance cannot be directly calculated due to the missing hydrological information."

**(5) Reviewer: P3 L4. "as an estimation model"**

**Answer:** The modification is done in the revised version of the manuscript.

> P3-L6: "This is achieved by replacing CCA in the prior analysis of hydrological variables by Projection Pursuit Regression (PPR), a nonparametric regression method recently considered as an estimation model in RFFA (Durocher et al., 2015)."

**(6) Reviewer: P9 L4. Please, define NH in equation 14.**

**Answer:** The authors agree that the notation is not clear. In Eq. (14), NH corresponds to the product of the variables N and H, which are defined. $N$ corresponds to the number of gauged sites in the neighborhoods and $H$ is the heterogeneity measure in Eq. (13) calculated on all $n$ available gauged sites.

In the revised version of the manuscript a product symbol is added to Eq. (14) to clarify that it is the product of two separate variables:

$$\text{AHM} = \frac{1}{N \cdot H} \sum_{j=1}^{n} H_{(j)}$$

**(7) Reviewer: P9 L14 – 15. How can a regression model fitted on two different neighborhoods, for the same target location, obtain identical values?**

**Answer:** A simple illustration would be to consider sites with value: {1,2,3,4,5}. Two delineation methods lead to the group {1,3,5} and {2,3,4}. Here, both groups have predicted value 3 as their mean, but the first group has a variance of 4 and the second has a variance of 1.

The following sentence will be modified in the revised version of the manuscript to clarify that "identical values" stands for "similar predicted values":

> P10-L3, "Notice that a regression model fitted on two different neighborhoods (for the same target location) can lead to very similar predictions, but with different levels of variance."

**(8) Reviewer: P10 L2. I don't believe that 15 years of data are enough to get statistically reliable results, why did authors choose 15 years as the minimum time series used in the study.**

**Answer:** This choice has not been made in the present study. We are using in this study a database that has been used in a number of previous studies. This allows to compare the results with those obtained with other methods that are now commonly accepted. This is explained in the revised manuscript:

> P10-L20, "Only a brief description of the data and the at-site frequency analysis is provided since the elements were presented in details in previous studies (e.g., Chokmani and Ouarda, 2004)."

Notice also that 15 years is a minimum. Most time series are much longer. The average length is added to the revised version of the manuscript to highlight this point:

> P10-L19, "Each site has at least 15 years of data available, with an average length of 31 years."

**(9) Reviewer: P10 L3. I think you should have at least a map showing the locations of the selected stations in the case study (Quebec).**

**Answer:** The authors agree with the reviewer and a new map (Figure 2) is added to the revised version of the manuscript:

> P10-L18, "The analysis is performed on 151 sites located in Southern Quebec, Canada, which are presented in Figure 2."

**(10) Reviewer: P10 L7. Using the maximum likelihood for parameter estimation with small time series (e.g., 15 years) may cause convergence problems, I would recommend using L-moments instead.**

**Answer:** As mentioned in the minor comment (8), the at-site frequency analysis was performed and validated in previous studies, where the necessary precautions were taken to ensure reliable estimates. Moreover, notice the use of the terms "including" and "In general" in the following sentences of the revised manuscript to underline that only a brief description of the full methodology is provided:

> P10-L23, "The at-site distributions are selected among several families including: generalized extreme values (GEV), Pearson type III (P3), generalized logistic (GLO) and log-normal with 3 parameters (LN3). In general, the estimation of the at-site distribution was achieved by maximum likelihood and the final choices of distributions are based on the Akaike information criterion."

**(11) Reviewer: P11 L16. What does HYB denote for in "RVN-HYB"?**

**Answer:** The acronym "HYB" stands for hybrid, because hydrological and physiographical variables are used as reference variables. The following sentences will be added to the revised version of the manuscript to clarify this point:

> P11-L4, "The first group is based on L-moments only and the second is based on the combination of L-moments and site-characteristics. The acronym LM for L-moment and HYB for Hybrid are used to identify the two groups."

**(12) Reviewer: P11 L19. "One of the objectives of RFA is to identify a proper family of distributions from regional information" This is not an objective of the RFA. I suggest to write one of the main steps.**

**Answer:** In the revised version of the manuscript the sentence has been modified as follows:

> P12-12, "Moreover, it is necessary to identify a proper family of distributions from regional information, which is achieved here by analyzing the distribution of the gauged sites inside the neighborhoods."

**(13) Reviewer: Please, define here the Q(r) as the regional quantile. The authors defined the Q(r) on P11 L29 but it still needed to be defined immediately after the equation in P11 L23.**

**Answer:** The author agrees with the reviewer. The change indicated by the reviewer is done in the revised version of the manuscript:

> P12-L18, "In this model, the regional quantile $Q_i(r) = \mu_i Q(r)$ corresponding to a return period $r$ at a target location $i$, where $\mu_i$ is the index-flood."

**(14) Reviewer: The second part of the title of figure 3: (b), (c), and (d) Regional L-moments based on the 15 nearest gauged sites for 3 selected target locations.**

The correction is made in the revised version of the manuscript. The authors reiterate their thanks to the reviewer.

[revised manuscript text omitted]

---

## Referee Report (RR1)

**General Comments:**

The present manuscript investigates the utilization of hydrological information in Regional Frequency Analysis (RFA) in order to improve homogenous properties of neighborhoods and then improve regional flood estimation. I think that the contribution of the study is relevant, and the authors have adequately responded to almost all the comments of the reviewers. However, I still have one "unfinished" major comment and three specific comments.

**Major comments:**

***The original comment:*** Since, homogeneity tests (*e.g.* Hosking and Wallis 1997) are generally based on hydrological variables (*e.g.*, L1, LCV), these variables should not be used in delineating homogeneous regions. In other words, the same information should not be used for both delineating the homogeneous regions and testing the homogeneity of such regions.

***Authors:*** *the present methodology does not perform any homogeneity test. The criteria used for selecting the size of the neighborhood is the RRMSE and is based on cross- validation, which tends to optimize the prediction corresponding to a specific return period. Consequently, the L-moments are not the variables used in the calibration of the neighborhood.*

The authors replied to this comment that "*the present methodology does not perform any homogeneity test*". However, the comparative study in their manuscript based on four criteria including the AHM which depends on the heterogeneity measure (*H*). Hence, the proposed comparative methodology assesses the considered methods based on the homogeneity of the resulted delineated regions.

Surprisingly, the authors replied here that "*the L-moments are not the variables used in the calibration of the neighborhood*". Nevertheless, the proposed method (RVN) based on reference variables which mainly include the L-moments. Review P11 L3 "*Two initial groups of reference variables are considered. The first group is based on L-moments only and the second is based on the combination of L-moments and site-characteristics.*"

**Specific comments:**

1. P6 L21. "Calculate the distance between the reference variables". Again, the distance is between locations not between variables. The authors have already agreed that the formulation needs to be changed and distances remain between locations not between variables.

2. Figure 2. I would like to thank the authors for their respond to my recommendation of drawing a map of Quebec showing the location of the selected stations. However, the map should include more labels (*e.g*., Quebec, Atlantic Ocean, Hudson Bay….). Also, the style of the map looks very old (we should take advantage of the recent technology in map drawing). Please, see Figure 1 (Reprinted from Gado and Nguyen 2016, © ASCE) for a close example of a map of Quebec.

3. Figure 7. Thanks again for accepting my suggestion of using the Q-Q plot. However, I have recommended the Q-Q plot in order to compare the considered methods regarding the estimation of regional flood quantile, not to draw the Q-Q plot for every method separately which does not make sense in assessing the different methods. Please, see Figure 2 (Reprinted from Gado and Nguyen 2015, © ASCE) for an example to clarify my point.

[Figure]

Figure 1 (Reprinted from Gado and Nguyen 2016, © ASCE)

[Figure]

Figure 2 (Reprinted from Gado and Nguyen 2015, © ASCE)

**References**

Hosking, J.R.M., Wallis, J.R., 1997. Regional frequency analysis: an approach based on L-moments. Cambridge Univ Pr.

Gado, T. A., and Nguyen, V.T.V., 2015. Comparison of homogenous region delineation approaches for regional flood frequency analysis at ungauged sites. J. of Hydrol. Eng., 21(3), Doi: 10.1061/(ASCE)HE.1943-5584.0001312, 04015068.

Gado, T. A., and Nguyen, V.T.V., 2016. Regional Estimation of Floods for Ungauged Sites Using Partial Duration Series and Scaling Approach. J. of Hydrol. Eng., Doi: 10.1061/(ASCE)HE.1943-5584.0001439.

---

## Author Response (AR2)

**Delineation of homogenous regions using hydrological variables predicted by projection pursuit regression**

Martin Durocher, Fateh Chebana, and Taha B. M. J. Ouarda

Manuscript: hess-2016-123

Dear Editor,

Please find herewith the response to all comments made by the reviewers concerning the second revision of the manuscript "Delineation of homogenous regions using hydrological variables predicted by projection pursuit regression" for publication in Hydrology and Earth System Science (HESS).

The authors gratefully acknowledge the helpful comments that have contributed to the improvement of the paper. Detailed replies to these comments are provided below.

Should you have any questions or require further information concerning this revision, please contact me.

Sincerely,
Martin Durocher

**Reply to Editor comment**

**I confess that, even if you clarified many points, I still find the presentation cumbersome and the procedure difficult to follow and not described very clearly, as demonstrated also by the last doubts raised by Ref#2 in his major comment (it is not clear the use of the indexes presented in section 3.2, where both RRMSE and heterogeneity are presented, even if it is said that only RRMSE is used for calibrating the neighbourhood size…). It is now probably difficult to re-structure it completely (and after all, none of the Referees asked you to do so in their last reviews), but you should try to clarify such doubts (if the Referee - who has carefully read the paper more than once - still find some points dubious, it will be certainly worse for a 'regular' reader …). Therefore, I do hope you may find a way to further clarify the procedure; I would suggest adding a flow-diagram with all the steps and the variables and indexes used in the different phases.**

**Answer**: The authors are grateful to the editor for its comment. The suggestion of the diagram is incorporated to the manuscript. Further details of modification are provided in the answer of the reviewer 2 comments

**Reply to Reviewer 2 comments**

The authors provide hereafter the answers to the reviewer's comments.

**General comments**

**(1.1) The original comment: Since, homogeneity tests (e.g. Hosking and Wallis 1997) are generally based on hydrological variables (e.g., L1, LCV), these variables should not be used in delineating homogeneous regions. In other words, the same information should not be used for both delineating the homogeneous regions and testing the homogeneity of such regions.**

**Authors: the present methodology does not perform any homogeneity test. The criteria used for selecting the size of the neighborhood is the RRMSE and is based on cross- validation, which tends to optimize the prediction corresponding to a specific return period. Consequently, the L-moments are not the variables used in the calibration of the neighborhood.**

**The authors replied to this comment that "the present methodology does not perform any homogeneity test". However, the comparative study in their manuscript based on four criteria including the AHM which depends on the heterogeneity measure (H). Hence, the proposed comparative methodology assesses the considered methods based on the homogeneity of the resulted delineated regions."**

**Answer**: The authors apology if their response has not completely answered the previous comments. By its nature, the AHM criterion is, of course, biased in favor of the RVN method, which is made clearer in the revised manuscript:

P13 L18: "The AHM criterion indicates that the ROI and the CCA methods have in general lower heterogeneity than the whole dataset, but are largely outperformed by the RVN-LM and RVN-HYB methods especially for smaller neighborhoods. This is not surprising as the RVN-LM and RVN-HYB pool

together sites with similar L-moments, but this quantifies the intuitive assumption that the regional LCV is calculated with less uncertainty when the L-moments are directly considered instead of other reference variables."

The AHM criterion is used here simply as a general description of the global level of heterogeneity among all neighborhoods. The author agrees a comparison based uniquely on the AHM would be unfair. On the other hand, the RRMSE criterion is objective and for this reason it is used in the calibration. Although, the RFFA of the present case study could have been performed completely without reporting the AHM criterion, the authors believe that this criterion provides an interesting indication that the neighborhoods delineated using ROI do not lead to the level of hydrological similarity that one may expect. The sentences below are already part of the manuscript and show the poor level hydrological similarity among neighborhoods resulting from the ROI method:

P13 L23 :"In particular, the AHM of the ROI method is 72.8% with the optimal neighborhood size of 30. In comparison, the AHM of the RVN-LM method is 14.5% with the optimal neighborhood size of 29 sites, which is considerably lower. Figure 6c shows that the AHM criterion of the RVM-LM method does not reach a similar level to the ROI method until using as much as 120 sites. These results indicate that even for relatively small neighborhoods, the ROI method identifies regions that are only slightly less hydrologically heterogeneous than all sites pooled together. This suggests that, in the present case study, the ROI method has difficulties identifying sites that are similar to the target site in terms of LCV."

**(1.2) Surprisingly, the authors replied here that "the L-moments are not the variables used in the calibration of the neighborhood". Nevertheless, the proposed method (RVN) based on reference variables which mainly include the L-moments. Review P11 L3 "Two initial groups of reference variables are considered. The first group is based on L-moments only and the second is based on the combination of L-moments and site-characteristics.**

**Answer:** By the previous reply, the authors simply meant that the calibration is made according to the RRMSE, which is computed from the flood quantile and not the L-moments. As suggested by the editor, a diagram of the whole procedure is added in Figure 1 to clarify the process of selecting the reference variables. This diagram provides a better understanding of the backward stepwise selection procedure and the following explanations are also added.

P7 L1: "This selection procedure is more objective and depends on a performance criterion. In the present study the RRMSE criterion is chosen and will be described in section 3.2. The backward stepwise selection is illustrated in Figure 1 and consists to remove in turn each reference variable temporarily from the model and to perform the remaining steps (2-6) in order to compute the RRMSE. Therefore, the reference variable whose removing leads to the best RRMSE, is permanently removed. The process is repeated until all reference variables cannot be removed without altering the RRMSE."

The following sentences are also modified in the revised version of the manuscript and in particular to respond to the reviewer comment:

P11 L6 "Two initial groups of reference variables are considered and updated by backward stepwise selection. The first group is based on L-moments only and the second is based on the combination of L-moments and site-characteristics."

P12 L5: "Due to its poor fit, LSK may not be a proper reference variable for the delineation step. To validate this assumption, the neighborhoods are formed with and without using LSK and the rest of the analysis is carried out for both scenarios. Based on the RRMSE criterion, LSK must be maintained as it is associated to better predictive performances. This strategy is part of the backward stepwise selection procedure as described in section 3.1. Overall, it leads to discarding LKT and to maintaining L1, LCV and LSK. The second group of reference variables contains both the L-moments and the site characteristics. As with the first group, backward stepwise selection is performed and the final reference variables are: BV, PLAC, LCV and LSK. In order to distinguish the two groups of reference variables, RVN-LM will designate the first group with the L-moments only and RVN-HYB will designate the second group with both the L-moments and the site characteristics."

**Specific comments**

**(1) P6 L21. "Calculate the distance between the reference variables". Again, the distance is between locations not between variables. The authors have already agreed that the formulation needs to be changed and distances remain between locations not between variables**

**Answer**:  The authors agree to rewrite the sentence.

P6 L24 : "Calculate the distance between sites"

**(2) Figure 2. I would like to thank the authors for their respond to my recommendation of drawing a map of Quebec showing the location of the selected stations. However, the map should include more labels (e.g., Quebec, Atlantic Ocean, Hudson Bay….). Also, the style of the map looks very old (we should take advantage of the recent technology in map drawing).**

**Answer**:  The authors have added the labels to the map to improve its understanding as suggested by the reviewer. The authors can assure the reviewer that the map in Figure 2 is produce in R using the authors own codes and that the information presented is up to date.  Although this approach does not use the most recent GIS tools, it does provide a clear representation of the studied region.

**(3) Figure 7. Thanks again for accepting my suggestion of using the Q-Q plot. However, I have recommended the Q-Q plot in order to compare the considered methods regarding the estimation of regional flood quantile, not to draw the Q-Q plot for every method separately which does not make sense in assessing the different methods.**

Answer: The authors have made the changes in the QQ-plot as suggested by the reviewer for the comparison of pairs of methods. The result is provided in Figure 9 of the revised manuscript, with the following explanations:

[revised manuscript text omitted]